# FLOW MATCHING WITH GAUSSIAN PROCESS PRIORS FOR PROBABILISTIC TIME SERIES FORECASTING

**Marcel Kollovieh**[1,2,3]    **Marten Lienen**[1,2]    **David Lüdke**[1,2]

**Leo Schwinn**[1,2]    **Stephan Günnemann**[1,2,3]

[1] School of Computation, Information and Technology, Technical University of Munich
[2] Munich Data Science Institute    [3] Munich Center for Machine Learning

Correspondence to: `m.kollovieh@tum.de`

## ABSTRACT

Recent advancements in generative modeling, particularly diffusion models, have opened new directions for time series modeling, achieving state-of-the-art performance in forecasting and synthesis. However, the reliance of diffusion-based models on a simple, fixed prior complicates the generative process since the data and prior distributions differ significantly. We introduce TSFlow, a conditional flow matching (CFM) model for time series combining Gaussian processes, optimal transport paths, and data-dependent prior distributions. By incorporating (conditional) Gaussian processes, TSFlow aligns the prior distribution more closely with the temporal structure of the data, enhancing both unconditional and conditional generation. Furthermore, we propose conditional prior sampling to enable probabilistic forecasting with an unconditionally trained model. In our experimental evaluation on eight real-world datasets, we demonstrate the generative capabilities of TSFlow, producing high-quality unconditional samples. Finally, we show that both conditionally and unconditionally trained models achieve competitive results across multiple forecasting benchmarks.

## 1 INTRODUCTION

Diffusion (Sohl-Dickstein et al., 2015; Ho et al., 2020) and score-based generative models (Song et al., 2020) have demonstrated strong performance in time series analysis, effectively capturing the distribution of time series data in both conditional (Rasul et al., 2021; Tashiro et al., 2021; Biloš et al., 2023; Alcaraz & Strodthoff, 2022) and unconditional (Kollovieh et al., 2023) settings. However, these models typically transform non-i.i.d. distributions of time series data into a simple isotropic Gaussian prior by iteratively adding noise leading to long and complex paths. This can hinder the generative process and potentially limit the models' performance.

Conditional Flow Matching (CFM) (Lipman et al., 2022; Tong et al., 2023) provides an efficient alternative to diffusion models and simplifies trajectories by constructing probability paths based on conditional optimal transport. The model is trained by regressing the flow fields of these paths and can accommodate arbitrary prior distributions. Despite its advantages, the application of CFM to time series forecasting remains unexplored.

In this work, we simplify the trajectories of generative time series modeling through *informed priors*, specifically using Gaussian Processes (GPs) as prior distributions within the CFM framework. We align the prior distributions with the underlying temporal dynamics of real-world data via optimal transport couplings. This approach not only reduces the complexity of the learned transformations but also enhances the model's performance in conditional and unconditional generation tasks. Furthermore, we demonstrate how both conditionally and unconditionally trained models can be leveraged for probabilistic forecasting. We propose *conditional prior sampling* and demonstrate how to use guidance (Dhariwal & Nichol, 2021) to bridge the gap between conditional and unconditional generation, allowing for flexible application of unconditional models without the need for specialized conditional training procedures.

Our empirical evaluations show that using conditional GPs as priors improves generative modeling and forecasting performance, surpassing various baselines from different frameworks on multiple datasets. This validates the effectiveness of our approach in simplifying the generative problem and enhancing model versatility.

Our *key contributions* are summarized as follows:

- We introduce TSFlow, a novel generative model for time series that incorporates conditional Gaussian Processes as informed prior distributions within the Conditional Flow Matching framework.

- We demonstrate how these priors align with the data distribution and improve the unconditional generative performance.

- We show how to utilize both conditionally trained and unconditionally trained models for forecasting tasks, enhancing the flexibility and applicability of our approach.

- Through extensive empirical evaluation, we validate the effectiveness of TSFlow, achieving competitive performance and outperforming existing methods on several benchmark datasets.

## 2 BACKGROUND: CONDITIONAL FLOW MATCHING

Conditional Flow Matching (CFM) was recently introduced as a framework for generative modeling by learning a flow field to transform one distribution into another (Lipman et al., 2022). The learned flow field then yields a transformation $\phi_1$ that maps a sample $\mathbf{x}_0 \sim q_0$ from a prior distribution $q_0$, e.g., an isotropic Gaussian, to a transformed sample $\phi_1(\mathbf{x}_0)$ that follows the data distribution $q_1$, i.e., $\phi_1(\mathbf{x}_0) \sim q_1$. The flow field $\phi_t$ is parametrized through a vector field $u_{\boldsymbol{\theta}}$, which is learned in a simple regression task. In the following, we give a short exposition of flow matching. For a thorough introduction to flow matching and diffusion, we refer the reader to Luo (2022); Lipman et al. (2022).

### 2.1 PROBABILITY FLOWS

The flow $\phi_t(\mathbf{x})$ describes the path of a sample $\mathbf{x}$ following the time-dependent vector field $u_t(\mathbf{x})$ : $[0, 1] \times \mathbb{R}^d \to \mathbb{R}^d$ and is itself the solution to the ordinary differential equation

$$\mathrm{d}\phi_t(\mathbf{x}) = u_t(\phi_t(\mathbf{x}))\, \mathrm{d}t, \qquad \phi_0(\mathbf{x}) = \mathbf{x}_0. \tag{1}$$

For a given density $p_0$, $\phi_t$ induces a time-dependent density called a probability path via the push forward operator $p_t(\mathbf{x}) \coloneqq ([\phi_t]_* p_0)(\mathbf{x}) = p_0(\phi_t^{-1}(\mathbf{x})) \det[\mathrm{d}\phi_t^{-1}/\mathrm{d}x(\mathbf{x})]$. Our goal is to find a simple, i.e., short and straight, probability path $p_t$ whose boundary probabilities align with our prior and data distribution, i.e., $p_0 \approx q_0$ and $p_1 \approx q_1$.

If we now consider two flows $\phi_t$ and $\phi_t'$, their induced probability paths $p_t$ and $p_t'$ will be equal if their flow fields $u_t$ and $u_t'$ are equal. Consequently, we can train a generative model for a data distribution $q_1$ by matching a model $u_{\boldsymbol{\theta}}$ to a flow field $u_t$ corresponding to a probability path between a prior distribution $q_0$ and $q_1$.

Lipman et al. (2022) propose to model the marginal probability paths as a mixture of conditional paths, i.e.,

$$p_t(\mathbf{x}) = \int p_t(\mathbf{x} \mid \mathbf{z}) q(\mathbf{z}) \mathrm{d}\mathbf{z}. \tag{2}$$

More specifically, they choose low-variance Gaussian paths centered on each data point $\mathbf{x}_1$, i.e., $p_t(\mathbf{x} \mid \mathbf{x}_1) = \mathcal{N}(t\mathbf{x}_1, (1 - (1 - \sigma_{\min}^2)t)\mathbf{I})$, with $\mathbf{z} = \{\mathbf{x}_1\}$ and $q = q_1$, which result in an isotropic prior distribution, i.e., $q_0 = \mathcal{N}(\mathbf{0}, \mathbf{I})$. These conditional probability paths have a closed form for their conditional flow field $u_t(\mathbf{x} \mid \mathbf{x}_1)$. Furthermore, they show that the marginal flow field arising from these conditional flow fields generates the approximate data distribution $q_1$ and derive the Conditional Flow Matching objective:

$$\mathcal{L}_{\mathrm{CFM}}(\boldsymbol{\theta}) = \mathbb{E}_{t \sim \mathcal{U}[0,1], \mathbf{x}_1 \sim q_1, \mathbf{x} \sim p_t(\mathbf{x}|\mathbf{x}_1)} \|u_{\boldsymbol{\theta}}(t, \mathbf{x}) - u_t(\mathbf{x} \mid \mathbf{x}_1)\|^2. \tag{3}$$

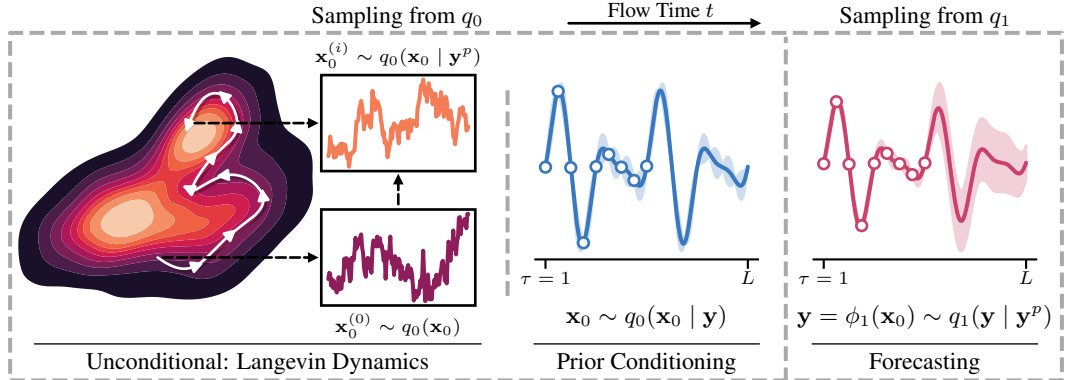

Figure 1: Overview of our proposed model TSFlow. To perform forecasting, we first sample $\mathbf{x}_0$ conditioned on our observation $\mathbf{y}^p$ from $q_0$. In an unconditional setting, we do this via Langevin dynamics (see Sec. 3.1.2), while in the conditional model, we can do this by directly using a conditional prior, e.g., a Gaussian process regression (see Sec. 3.2). Given $\mathbf{x}_0$, we can now sample from $q_1$ by solving its corresponding ODE (see Eq. (1)).

## 2.2 COUPLINGS

Tong et al. (2023) generalize Conditional Flow Matching to encapsulate arbitrary source distributions. This is achieved by including not only samples from the target distribution but also from the source into the conditioning of the probability paths, i.e., $\mathbf{z} = \{\mathbf{x}_0, \mathbf{x}_1\}$. More specifically, they choose

$$p_t(\mathbf{x} \mid \mathbf{x}_0, \mathbf{x}_1) = \mathcal{N}(t\mathbf{x}_1 + (1-t)\mathbf{x}_0, \sigma_{\min}^2 \mathbf{I}), \tag{4}$$

satisfying $p_0 = q_0$ and $p_1 = q_1$ for $\sigma_{\min}^2 \to 0$. The respective conditional vector field is simply the difference of $\mathbf{x}_1$ and $\mathbf{x}_0$, i.e., $u_t(\mathbf{x} \mid \mathbf{x}_0, \mathbf{x}_1) = \mathbf{x}_1 - \mathbf{x}_0$ and can be learned as a regression task:

$$\mathcal{L}(\boldsymbol{\theta}) = \mathbb{E}_{t \sim \mathcal{U}[0,1], (\mathbf{x}_0, \mathbf{x}_1) \sim q(\mathbf{x}_0, \mathbf{x}_1), \mathbf{x} \sim p_t(\mathbf{x} \mid \mathbf{x}_0, \mathbf{x}_1)} \left\| u_{\boldsymbol{\theta}}(t, \mathbf{x}) - u_t(\mathbf{x} \mid \mathbf{x}_0, \mathbf{x}_1) \right\|^2. \tag{5}$$

While the simple choice is to sample $\mathbf{x}_0$ and $\mathbf{x}_1$ independently, the framework allows for arbitrary joint distributions $q(\mathbf{x}_0, \mathbf{x}_1)$ with marginals $q_0$ and $q_1$.

## 2.3 MINI-BATCH OPTIMAL TRANSPORT

A natural choice for joint distributions is the optimal transport coupling between source and target distributions. In practice, however, finding the optimal transport map $\pi$ is only feasible for small datasets. Instead, Tong et al. propose a mini-batch variant of the algorithm. For each batch of data $\{\mathbf{x}_1^{(i)}\}_{i=1}^B$ seen during training, they sample $B$ points $\mathbf{x}_0^{(i)} \sim q_0$ and then compute $\pi$ between these batches. This makes finding $\pi$ computationally feasible while retaining the benefits of the optimal transport coupling empirically (Tong et al., 2023). Choosing $q$ as the optimal transport map $\pi$ between the prior and data distribution makes both training and sampling more efficient by lowering the variance of the objective and straightening the probability paths.

## 3 TSFLOW

In this section, we present our main contributions: **(1)** TSFlow, a novel flow matching model tailored for probabilistic time series forecasting (see Fig. 1). TSFlow supports both unconditional and conditional generation, i.e., generation without or with partially observed time series. **(2)** We explore non-i.i.d. prior distributions within the CFM framework and demonstrate that incorporating a Gaussian process improves unconditional generation by aligning the prior with the data distribution. Further, **(3)** we show how an unconditional model can be employed for forecasting by conditioning it during inference via Langevin dynamics and guidance techniques. Finally, **(4)** we describe how to use data-dependent prior distributions in the form of Gaussian process regression to train TSFlow for conditional forecasting directly.

**Problem Statement.** Consider a regularly sampled *univariate* time series $\mathbf{y} \in \mathbb{R}^L$ of length $L$ drawn from the data distribution $q_1(\mathbf{y})$ (corresponding to dimension $d$, as detailed in Sec. 2). We denote the time series as $\mathbf{y} = (\mathbf{y}^p \ \mathbf{y}^f)$, where $\mathbf{y}^p$ represents the observed past and $\mathbf{y}^f$ the unknown future we aim to predict. To avoid ambiguity with the flow-matching time parameter $t$, we introduce $\tau$ as the time index for positions within the time series data. Here, the flow-matching time $t$ parameterizes the progression along the transformation path from the prior to the target time series distribution, while $\tau$ is orthogonal and refers to specific time points in the observed data.

Our objective is to capture the conditional distribution $q_1(\mathbf{y} \mid \mathbf{y}^p)$ using a generative model $p_{\boldsymbol{\theta}}(\mathbf{y} \mid \mathbf{y}^p)$. By marginalizing over the noisy time series $\mathbf{x}_0$, we decompose the conditional model into a conditional prior and generative process:

$$p_{\boldsymbol{\theta}}(\mathbf{y} \mid \mathbf{y}^p) = \int p_{\boldsymbol{\theta}}(\mathbf{y} \mid \mathbf{x}_0, \mathbf{y}^p) \, q_0(\mathbf{x}_0 \mid \mathbf{y}^p) \mathrm{d}\mathbf{x}_0. \tag{6}$$

Choosing $q_0$ as an isotropic Gaussian (as in existing diffusion models) neglects to condition the prior on the observed past. In the following, we describe how TSFlow models both the conditional generative model and the conditional prior.

First, we introduce an unconditional model $p_{\boldsymbol{\theta}}(\mathbf{y})$ to capture the data distribution $q_1(\mathbf{y})$. During training, this model is unaware of any conditioning information, such as past observations. We then explain how to condition this model and its prior on past observations during inference, resulting in $p_{\boldsymbol{\theta}}(\mathbf{y} \mid \mathbf{y}^p)$ and $q_0(\mathbf{x}_0 \mid \mathbf{y}^p)$, respectively.

Then, we introduce a conditionally trained model that directly incorporates $\mathbf{y}^p$ to model the conditional distribution $p_{\boldsymbol{\theta}}(\mathbf{y} \mid \mathbf{y}^p)$. To simplify notation, we denote $\mathbf{x}_1$ as $\mathbf{y}$ in the following. Furthermore, we denote the noisy time series as $\mathbf{x}_0$, and its (temporal) past and future $\mathbf{x}_0^p$ and $\mathbf{x}_0^f$, respectively.

## 3.1 UNCONDITIONAL MODELING

---

Algorithm 1: Unconditional Training of TSFlow

---

1: **Input:** Prior distribution $q_0$, data distribution $q_1$, noise levels $\sigma_t$ network $u_{\boldsymbol{\theta}}$
2: **for** iteration $= 1, \dots$ **do**
3:     $\mathbf{x}_0 \sim q_0(\mathbf{x}_0); \ \mathbf{y} \sim q_1(\mathbf{y})$                    ▷ sample batches from the prior and dataset
4:     $(\mathbf{x}_0, \mathbf{y}) \leftarrow \mathrm{OT}(\mathbf{x}_0, \mathbf{y})$                    ▷ compute optimal transport mapping
5:     $t \sim \mathcal{U}(0, 1)$                    ▷ sample random time step $t$
6:     $\boldsymbol{\mu}_t \leftarrow t\mathbf{y} + (1 - t)\mathbf{x}_0$                    ▷ compute mean of $p_t$
7:     $\mathbf{x}_t \sim \mathcal{N}(\boldsymbol{\mu}_t, \sigma_t^2 \mathbf{I})$                    ▷ sample from $p_t(\cdot \mid \mathbf{z})$
8:     $\mathcal{L}(\boldsymbol{\theta}) \leftarrow \|u_{\boldsymbol{\theta}}(t, \mathbf{x}_t) - u_t(\mathbf{x}_t \mid \mathbf{z})\|^2$                    ▷ regress conditional vector field
9:     $\boldsymbol{\theta} \leftarrow \mathrm{Update}(\boldsymbol{\theta}, \nabla_{\boldsymbol{\theta}}\mathcal{L}(\boldsymbol{\theta}))$                    ▷ gradient step
10: **end for**
11: **Return:** $u_{\boldsymbol{\theta}}$

---

We begin by modeling the data distribution $q_1(\mathbf{y})$ using an unconditional CFM model $p_{\boldsymbol{\theta}}(\mathbf{y})$, which is optimized using mini-batch optimal transport couplings (see Sec. 2.3 and Alg. 1) and does not incorporate any conditioning information during training.

To better capture the temporal correlations in the time series $\mathbf{y}$, we explore Gaussian process priors instead of the default isotropic Gaussian distribution. By introducing temporal structure and aligning the prior distribution $q_0$ more closely with the data distribution $q_1$ via optimal transport couplings, we enhance the model's ability to learn temporal patterns (Sec. 3.1.1). Then, we describe how to condition the unconditional model on observed data $\mathbf{y}^p$ during inference by incorporating past observations into the generation process via Langevin dynamics and guidance, effectively transforming it into a conditional one (Sec. 3.1.2 and 3.1.3).

### 3.1.1 INFORMED PRIOR DISTRIBUTIONS

Previous work by Tong et al. (2023) has shown that pairing prior and data samples based on the optimal transport map accelerates training, reduces the number of Neural Function Evaluations (NFEs)

required during inference, and enhances the model's performance by shortening the conditional paths and straightening the learned velocity field.

We can further enhance this effect by choosing a domain-specific prior distribution $q_0$ closer to the data distribution $q_1$ than the standard isotropic Gaussian, yet similarly easy to specify and sample from. In the context of time series, we propose to employ Gaussian process (GP) priors $\mathcal{GP}(0, K)$ with a kernel function $K(\tau, \tau')$ (Rasmussen & Williams, 2005). Gaussian processes are well-suited for modeling time series data because they naturally capture temporal correlations. By choosing $K$, we can incorporate dataset-specific structures and temporal patterns into the prior without requiring an extensive training process.

We explore three kernel functions that reflect different types of data characteristics: squared exponential (SE), Ornstein-Uhlenbeck (OU), and periodic (PE) kernels, defined respectively as:

$$K_{\mathrm{SE}}(\tau, \tau') = \exp\Big(-\frac{d^2}{2\ell^2}\Big),\ K_{\mathrm{OU}}(\tau, \tau') = \exp\Big(-\frac{|d|}{\ell}\Big),\ \text{and } K_{\mathrm{PE}}(\tau, \tau') = \exp\Big(-\frac{2}{\ell^2}\sin^2(d)\Big),$$

where $d = \tau - \tau'$ and $\ell$ is a non-negative parameter that adjusts the length scale of the kernel.

The squared exponential kernel $K_{\mathrm{SE}}$ produces infinitely smooth samples, making it suitable for modeling time series with a high degree of smoothness. The Ornstein-Uhlenbeck kernel $K_{\mathrm{OU}}$ is closely related to Brownian motion and ideal for modeling data with a rougher structure. The periodic kernel $K_{\mathrm{PE}}$ captures repeating patterns in the data by modeling the covariance between points as a function of their periodic differences, making it suitable for time series with periodic behavior. We provide samples drawn from these priors in App. A.6.

### 3.1.2 CONDITIONAL PRIOR SAMPLING

To adapt the unconditional model for conditional generation post-training, we propose *conditional prior sampling*, which conditions the prior distribution as $q_0(\mathbf{x}_0 \mid \mathbf{y}^p)$. Given a sample $\mathbf{x}_0 \sim q_0(\mathbf{x}_0 \mid \mathbf{y}^p)$, we use it as the initial condition for our vector field to generate new samples from the distribution $\mathbf{y} \mid \mathbf{x}_0$. It is important to note that this only conditions the prior distribution, not the generation process itself, which we discuss in Sec. 3.1.3.

Given an observation $\mathbf{y}^p$, we aim to find a corresponding sample $\mathbf{x}_0$ from the prior distribution that aligns with it. Formally, this entails sampling from the distribution $q_0(\mathbf{x}_0 \mid \mathbf{y}^p)$. If we have access to its score function $\nabla_{\mathbf{x}_0} \log q_0(\mathbf{x}_0 \mid \mathbf{y}^p)$, we can sample from this distribution via Langevin dynamics:

$$\mathbf{x}_0^{(i+1)} = \mathbf{x}_0^{(i)} - \eta \nabla_{\mathbf{x}_0} \log q_0(\mathbf{x}_0^{(i)} \mid \mathbf{y}^p) + \sqrt{2\eta}\xi_i \text{ with } \xi_i \sim \mathcal{N}(\mathbf{0}, \mathbf{I}) \tag{7}$$

where $\eta$ is a fixed step size. With sufficient iterations, Eq. (7) converges to the conditional distribution (see Durmus & Moulines (2017) for a convergence analysis).

By applying Bayes' rule, we express the conditional score function as:

$$\nabla_{\mathbf{x}_0} \log q_0(\mathbf{x}_0 \mid \mathbf{y}^p) = \nabla_{\mathbf{x}_0} \log q_1(\mathbf{y}^p \mid \mathbf{x}_0) + \nabla_{\mathbf{x}_0} \log q_0(\mathbf{x}_0). \tag{8}$$

Here, the term $\nabla_{\mathbf{x}_0} \log q_1(\mathbf{y}^p \mid \mathbf{x}_0)$ guides $\mathbf{x}_0$ towards evolving into a sample (after solving the ODE) aligning with the observation $\mathbf{y}^p$, while the term $\nabla_{\mathbf{x}_0} \log q_0(\mathbf{x}_0)$ ensures adherence to the prior distribution's manifold.

As $q_0(\mathbf{x}_0)$ is a GP, we can compute its likelihood in closed form. However, the term $q_1(\mathbf{y}^p \mid \mathbf{x}_0)$ is unknown and requires more attention. We follow Kollovieh et al. (2023) and model it as an asymmetric Laplace distribution centered on the output of the flow $\phi_{\boldsymbol{\theta},1}$ learned by our model:

$$q_1(\mathbf{y}^p \mid \mathbf{x}_0) = \mathrm{ALD}(\mathbf{y}^p \mid \phi_{\boldsymbol{\theta},1}(\mathbf{x}_0), \kappa), \tag{9}$$

leading to the quantile loss after applying the logarithm with quantile $\kappa$. This aligns better with probabilistic forecasting, which is evaluated using distribution-based metrics. To accelerate the evaluation of the score function, we approximate the integration of the flow field in $\phi_{\boldsymbol{\theta},1}$ in Eq. (9) using a small number of Euler steps.

By dynamically choosing the number of iterations in the Langevin dynamics, we can control how strongly the sample should resemble the observation. We provide a pseudocode in Alg. 3.

### 3.1.3 GUIDED GENERATION

To perform conditional generation more effectively, we can employ guidance techniques (Dhariwal & Nichol, 2021; Kollovieh et al., 2023) to modify the score function of the model, directly conditioning the generation process on the observed data $\mathbf{y}^p$. Unlike the previous method, where we only conditioned the prior distribution, we now adjust the dynamics of the generation process itself.

We start by modeling the conditional score function using Bayes' rule, similar to Eq. (8):

$$\nabla_{\mathbf{x}_t} \log p_t(\mathbf{x}_t \mid \mathbf{y}^p) = \nabla_{\mathbf{x}_t} \log p_t(\mathbf{y}^p \mid \mathbf{x}_t) + \nabla_{\mathbf{x}_t} \log p_t(\mathbf{x}_t). \tag{10}$$

Here, the term $\log p_t(\mathbf{y}^p \mid \mathbf{x}_t)$ acts as a guidance term that steers the generation process toward producing samples consistent with the observed past $\mathbf{y}^p$.

To incorporate this guidance into the generation process, we adjust the vector field $u_{\boldsymbol{\theta}}$ and subtract the guidance score, scaled by a factor $s$:

$$\tilde{u}_{\boldsymbol{\theta}}(t, \mathbf{x}_t) = u_{\boldsymbol{\theta}}(t, \mathbf{x}_t) - s\sigma_t \nabla_{\mathbf{x}_t} \log p_t(\mathbf{y}^p \mid \mathbf{x}_t). \tag{11}$$

We provide a detailed derivation of $\tilde{u}_{\boldsymbol{\theta}}(\mathbf{x}_t, t)$ in App. C. By integrating this modified vector field $\tilde{u}_{\boldsymbol{\theta}}$ over time, we generate samples that are conditioned on the observed data $\mathbf{y}^p$. The parameter $s$ allows us to control the influence of the conditioning, with larger values of $s$ resulting in samples that more closely resemble the observed past. To model the guidance score, we again follow (Kollovieh et al., 2023) and use an asymmetric Laplace distribution centered around the output of the flow as before:

$$p_t(\mathbf{y}^p \mid \mathbf{x}_t) = \mathrm{ALD}(\mathbf{y}^p \mid \phi_{\boldsymbol{\theta},1}(\mathbf{x}_t), \kappa), \tag{12}$$

where $\kappa$ is a parameter controlling the asymmetry of the distribution.

By incorporating the guidance term into the vector field and appropriately modeling the guidance score, we effectively condition the generation process to produce forecasts consistent with the observed past, enhancing the model's ability to generate accurate and coherent time series predictions.

### 3.2 CONDITIONAL MODELING

TSFlow imposes additional structure on the prior distribution $q_0$ in the form of a Gaussian process, a non-parametric time series model. We build upon this and propose an alternative training and inference approach for conditional generation, i.e., forecasting, with TSFlow. Instead of only conditioning the sampling process on an observed sample $\mathbf{y}^p$, we additionally condition the prior distribution on the observed past data by factorizing $q_0(\mathbf{x}_0 \mid \mathbf{y}^p)$ with $q_0(\mathbf{x}_0^p \mid \mathbf{y}^p)\, q_0(\mathbf{x}_0^f \mid \mathbf{y}^p)$.

While Tong et al. (2023) couple $\mathbf{x}_0$ and $\mathbf{y}$ via the optimal transport map $\pi$ during training (Sec. 2.3), we exploit the GP prior and define the joint distribution as $q(\mathbf{x}_0, \mathbf{y}) = q_1(\mathbf{y})\, q_0(\mathbf{x}_0 \mid \mathbf{y}^p)$. We condition the model using equation Eq. (6) during inference. Consequently, our choice of joint distribution $q(\mathbf{x}_0, \mathbf{y})$ effectively trains the model with a conditional prior distribution. This aligns the training process with how the model is employed during inference and improves forecasting performance (see Sec. 4.2).

**Gaussian Process Regression.** Since $q_0(\mathbf{x}_0)$ is a GP, we can compute $q_0(\mathbf{x}_0 \mid \mathbf{y}^p)$ analytically. We begin by factorizing the conditional prior as:

$$q_0(\mathbf{x}_0 \mid \mathbf{y}^p) = q_0(\mathbf{x}_0^p \mid \mathbf{y}^p)\, q_0(\mathbf{x}_0^f \mid \mathbf{y}^p),$$

which follows by assuming independence of $\mathbf{x}^p$ and $\mathbf{x}^f$ given $\mathbf{y}^p$. We choose $q_0(\mathbf{x}_0^p \mid \mathbf{y}^p) = \delta(\mathbf{x}_0^p - \mathbf{y}^p)$ to retain the information from the observation. For the unobserved future part, we employ GPR to model the conditional distribution (Rasmussen & Williams, 2005). Specifically,

$$q_0(\mathbf{x}_0^f \mid \mathbf{y}^p) = \mathcal{N}(\mathbf{x}_0^f \mid \boldsymbol{\mu}_{f|p}, \boldsymbol{\Sigma}_{f|p}), \tag{13}$$

with

$$\boldsymbol{\mu}_{f|p} = \boldsymbol{\Sigma}_{fp}\boldsymbol{\Sigma}_{pp}^{-1}\mathbf{y}^p \quad \text{and} \quad \boldsymbol{\Sigma}_{f|p} = \boldsymbol{\Sigma}_{ff} - \boldsymbol{\Sigma}_{fp}\boldsymbol{\Sigma}_{pp}^{-1}\boldsymbol{\Sigma}_{pf}, \tag{14}$$

where $\boldsymbol{\Sigma}_{ff} = K(f, f)$, $\boldsymbol{\Sigma}_{fp} = K(f, p)$, $\boldsymbol{\Sigma}_{pf}, \boldsymbol{\Sigma}_{pp} = K(p, p)$ correspond to covariance matrices computed using the kernel function $K$.

Here, $K(f, f)$ represents the covariance matrix among all future points in the series, indicating how each point in the future relates to others and similarly for $K(p, p)$ among past points. $K(f, p)$ describes the covariance between future and past points, illustrating how future values are expected to relate to the observed past. Conversely, $K(p, f)$ is the transpose of $K(f, p)$, reflecting the same relationships but from the perspective of past points relating to future points.

Rather than learning the transformation from an isotropic Gaussian to the data distribution, TSFlow leverages an informed prior simplifying the modeling process by utilizing structured, relevant historical data. Note that this approach introduces only minimal computational overhead, as the covariance matrices are computed just once, and only a single vector-matrix multiplication is required for each new instance. We provide the training algorithm in Alg. 2.

## 4 EXPERIMENTS

In this section, we present our empirical results and compare TSFlow against various baselines using real-world datasets. Our primary objectives are threefold: First, to determine whether the generative capabilities are competitive to other generative frameworks. Second, to investigate the impact of different prior distributions on the performance. Third, to evaluate how TSFlow compares to other models in probabilistic forecasting. Additionally, we aim to explore how conditional prior sampling of the unconditional version of TSFlow fares against its conditionally trained counterpart.

**Datasets.** We conduct experiments on eight *univariate* time series datasets from various domains and with different frequencies from GluonTS (Alexandrov et al., 2020). Specifically, we use the datasets Electricity (Dheeru & Taniskidou, 2017), Exchange (Lai et al., 2018), KDDCup (Godahewa et al., 2021), M4-Hourly (Makridakis et al., 2020), Solar (Lai et al., 2018), Traffic (Dheeru & Taniskidou, 2017), UberTLC-Hourly (FiveThirtyEight, 2016), and Wikipedia (Gasthaus et al., 2019). We provide further details about the datasets in App. A.1.

**Baselines.** To benchmark the unconditional performance of TSFlow, our experiments include TimeVAE (Desai et al., 2021) and TSDiff (Kollovieh et al., 2023), representing different types of generative models. To assess the forecasting performance of TSFlow, i.e., conditional generation, we compare against various established time series forecasting methods. This includes traditional statistical methods such as Seasonal Naive (SN), AutoARIMA, and AutoETS (Hyndman et al., 2008). In the domain of deep learning, we evaluate TSFlow against established models including DLinear (Zeng et al., 2023), DeepAR (Salinas et al., 2020), TFT (Temporal Fusion Transformers) (Lim et al., 2021), WaveNet (Oord et al., 2016), and PatchTST (Nie et al., 2022). Finally, we extend our comparison to include the four diffusion-based approaches CSDI (Tashiro et al., 2021), TSDiff (Kollovieh et al., 2023), SSSD (Alcaraz & Strodthoff, 2022), and (Biloš et al., 2023) which represent generative models in probabilistic time series forecasting. We provide more information in App. A.4.

**Evaluation Metrics.** To assess the generative capabilities of TSFlow in an unconditional setting, we calculate the 2-Wasserstein distance between the synthetic and real samples. Specifically, we generate 10,000 samples using our defined context length for this evaluation. Additionally, to compare the quality of the synthetic samples in a downstream task, we employ the *Linear Predictive Score* (LPS) (Kollovieh et al., 2023), which is defined as the (real) test CRPS of a linear regression model trained on synthetic samples (see App. A.5 for more details).

To evaluate the probabilistic forecasts, we use the *Continuous Ranked Probability Score* (CRPS) (Gneiting & Raftery, 2007), defined as

$$\text{CRPS}(F^{-1}, y) = \int_0^1 2\Lambda_\kappa(F^{-1}(\kappa), y)\, \mathrm{d}\kappa,$$

where $\Lambda_\kappa(q, y) = (\kappa - \mathbb{1}_{\{y < q\}})(y - q)$ represents the pinball loss at a specific quantile level $\kappa$ and $F$ is the cumulative distribution function of the forecast. CRPS is a proper scoring function that reaches its minimum when the forecast distribution $F$ coincides with the target value $y$. As computing the integral is generally not feasible, we follow previous works (Rasul et al., 2021; Kollovieh et al., 2023; Rasul et al., 2020) and approximate the CRPS using nine uniformly distributed quantile levels $\{0.1, 0.2, \ldots, 0.9\}$. The randomized methods approximate $F$ using 100 samples.

**Practical Considerations.** TSFlow uses a DiffWave (Kong et al., 2020) architecture with S4 layers (Gu et al., 2021) similar to previous works (Kollovieh et al., 2023; Alcaraz & Strodthoff, 2022). We depict the architecture in Fig. 3. The conditional model (see Sec. 3.2) gets an additional input $\mathbf{c}$, which contains $\mathbf{y}^p$ and a binary observation mask indicating which steps are observed. We train the model using Adam (Kingma & Ba, 2014) with a learning rate of $10^{-3}$ and gradient clipping set to $0.5$ and use a standard Euler ODE solver. Furthermore, we do not fit the Gaussian processes to the corresponding datasets to keep the prior distributions simple. Finally, we report the mean and standard deviations of five random seeds for all experiments to ensure reproducibility.

## 4.1 UNCONDITIONAL GENERATION

We test the unconditional generative capability of TSFlow in three experiments. First, we investigate how well the synthetic samples align with the real data. Then, we train a downstream model on synthetic data and compare it to the performance of the real data. Finally, we test *conditional prior sampling* (see Sec. 3.1.2) to condition TSFlow during inference time.

### 4.1.1 GENERATIVE CAPABILITIES

We follow Tong et al. (2023) and compute the 2-Wasserstein distance of 10,000 synthetic and real samples for TSFlow using different priors to investigate whether the optimal transport problem is simplified with non-homoscedastic distributions while keeping confounders constant. We report the results in Tab. 1.

Table 1: 2-Wasserstein distance between real and synthetic samples on eight real-world datasets for different generative models and prior distributions. Best scores in **bold**, second best underlined.

| | Method | NFE($\downarrow$) | Electr. | Exchange | KDDCup | M4 (H) | Solar | Traffic | UberTLC | Wiki2000 |
|---|---|---|---|---|---|---|---|---|---|---|
| | TimeVAE | 1 | $3.201_{\pm 0.08}$ | $0.034_{\pm 0.005}$ | $39.975_{\pm 13.239}$ | $7.071_{\pm 0.096}$ | $7.537_{\pm 0.092}$ | $9.626_{\pm 0.241}$ | $79.35_{\pm 16.065}$ | $218.764_{\pm 2.021}$ |
| | TSDiff | 100 | $3.112_{\pm 0.32}$ | $0.057_{\pm 0.035}$ | $29.556_{\pm 1.445}$ | $6.095_{\pm 0.292}$ | $5.422_{\pm 1.173}$ | $7.228_{\pm 0.128}$ | $67.281_{\pm 6.992}$ | $214.981_{\pm 20.16}$ |
| TSFlow | Isotropic | 4 | $2.929_{\pm 0.051}$ | $0.027_{\pm 0.003}$ | $27.264_{\pm 0.232}$ | $6.539_{\pm 0.073}$ | $5.193_{\pm 0.097}$ | $7.515_{\pm 0.081}$ | $62.019_{\pm 0.587}$ | $211.369_{\pm 1.545}$ |
| | OU | 4 | $2.696_{\pm 0.068}$ | $0.026_{\pm 0.004}$ | $\mathbf{23.762}_{\pm 0.113}$ | $6.509_{\pm 0.051}$ | $4.564_{\pm 0.075}$ | $7.283_{\pm 0.098}$ | $\mathbf{60.338}_{\pm 0.075}$ | $203.970_{\pm 7.638}$ |
| | SE | 4 | $2.704_{\pm 0.056}$ | $0.027_{\pm 0.003}$ | $\underline{23.786}_{\pm 0.041}$ | $6.543_{\pm 0.049}$ | $\underline{4.388}_{\pm 0.016}$ | $7.375_{\pm 0.082}$ | $\underline{60.452}_{\pm 0.388}$ | $205.748_{\pm 4.614}$ |
| | PE | 4 | $2.695_{\pm 0.033}$ | $0.027_{\pm 0.003}$ | $27.029_{\pm 0.239}$ | $6.704_{\pm 0.070}$ | $4.492_{\pm 0.045}$ | $7.450_{\pm 0.021}$ | $60.951_{\pm 0.373}$ | $205.856_{\pm 4.834}$ |
| TSFlow | Isotropic | 16 | $2.752_{\pm 0.024}$ | $\underline{0.025}_{\pm 0.003}$ | $26.332_{\pm 0.223}$ | $\underline{5.910}_{\pm 0.052}$ | $4.495_{\pm 0.055}$ | $7.166_{\pm 0.086}$ | $65.939_{\pm 0.833}$ | $204.269_{\pm 1.712}$ |
| | OU | 16 | $2.635_{\pm 0.045}$ | $\mathbf{0.023}_{\pm 0.003}$ | $25.222_{\pm 0.259}$ | $\mathbf{5.893}_{\pm 0.083}$ | $4.402_{\pm 0.065}$ | $\mathbf{6.990}_{\pm 0.039}$ | $64.871_{\pm 1.083}$ | $\underline{200.808}_{\pm 6.350}$ |
| | SE | 16 | $\underline{2.625}_{\pm 0.052}$ | $\mathbf{0.023}_{\pm 0.003}$ | $25.223_{\pm 0.340}$ | $5.937_{\pm 0.064}$ | $4.429_{\pm 0.064}$ | $\underline{6.991}_{\pm 0.054}$ | $64.172_{\pm 0.690}$ | $\mathbf{199.880}_{\pm 7.743}$ |
| | PE | 16 | $\mathbf{2.542}_{\pm 0.049}$ | $\mathbf{0.023}_{\pm 0.003}$ | $25.472_{\pm 0.102}$ | $5.992_{\pm 0.061}$ | $\mathbf{4.362}_{\pm 0.032}$ | $7.051_{\pm 0.065}$ | $64.102_{\pm 0.723}$ | $200.829_{\pm 8.673}$ |

We observe that the three domain-specific Gaussian process priors outperform the isotropic prior, except on the datasets Exchange and UberTLC. Furthermore, on most datasets, the non-isotropic prior distributions with 4 NFEs are competitive with the isotropic prior with 16 NFEs. All variations of TSFlow consistently outperform TimeVAE, demonstrating its strong generative capabilities. TSFlow also outperforms TSDiff with 16 NFEs and remains competitive when reducing to 4 NFEs. These results align with our observations in Fig. 6 and demonstrate that choosing a suitable prior distribution improves unconditional generation.

### 4.1.2 TRAINING DOWNSTREAM MODELS

In addition to the 2-Wasserstein distances, we compute the Linear Predictive Score (LPS), which measures the performance of a linear regression model trained on synthetic samples and evaluated on the real test set. We present these results in Tab. 2.

Table 2: LPS for different generative models on eight real-world datasets. Best scores in **bold**, second best underlined.

| | Method | Electr. | Exchange | KDDCup | M4 (H) | Solar | Traffic | UberTLC | Wiki2000 |
|---|---|---|---|---|---|---|---|---|---|
| | TimeVAE | $0.161_{\pm 0.013}$ | $0.012_{\pm 0.000}$ | $5.853_{\pm 0.781}$ | $0.053_{\pm 0.007}$ | $0.864_{\pm 0.058}$ | $0.504_{\pm 0.026}$ | $0.754_{\pm 0.092}$ | $0.891_{\pm 0.076}$ |
| | TSDiff | $0.094_{\pm 0.007}$ | $\mathbf{0.010}_{\pm 0.000}$ | $\mathbf{0.651}_{\pm 0.036}$ | $0.042_{\pm 0.014}$ | $0.634_{\pm 0.009}$ | $0.235_{\pm 0.006}$ | $0.392_{\pm 0.011}$ | $0.369_{\pm 0.027}$ |
| TSFlow | Isotropic | $\underline{0.089}_{\pm 0.002}$ | $\mathbf{0.010}_{\pm 0.000}$ | $\underline{0.684}_{\pm 0.047}$ | $\underline{0.031}_{\pm 0.001}$ | $\underline{0.607}_{\pm 0.003}$ | $\underline{0.232}_{\pm 0.003}$ | $0.360_{\pm 0.003}$ | $\underline{0.349}_{\pm 0.005}$ |
| | OU | $0.096_{\pm 0.007}$ | $\underline{0.011}_{\pm 0.000}$ | $0.722_{\pm 0.046}$ | $0.032_{\pm 0.001}$ | $0.616_{\pm 0.002}$ | $0.237_{\pm 0.001}$ | $0.352_{\pm 0.007}$ | $0.361_{\pm 0.003}$ |
| | SE | $0.102_{\pm 0.007}$ | $\underline{0.011}_{\pm 0.000}$ | $0.726_{\pm 0.056}$ | $0.032_{\pm 0.001}$ | $0.621_{\pm 0.001}$ | $0.241_{\pm 0.003}$ | $\underline{0.351}_{\pm 0.008}$ | $0.365_{\pm 0.003}$ |
| | PE | $\mathbf{0.083}_{\pm 0.006}$ | $0.012_{\pm 0.000}$ | $0.900_{\pm 0.082}$ | $\mathbf{0.029}_{\pm 0.001}$ | $\mathbf{0.590}_{\pm 0.003}$ | $\mathbf{0.225}_{\pm 0.003}$ | $\mathbf{0.320}_{\pm 0.006}$ | $\mathbf{0.339}_{\pm 0.005}$ |

As we observe, all variations of TSFlow consistently outperform TimeVAE and, on four out of eight datasets, also surpass TSDiff. The periodic prior outperforms the other priors, achieving the best scores on six datasets. TSDiff only marginally outperforms TSFlow on the Exchange and KDDCup datasets.

## 4.2 PROBABILISTIC FORECASTING

Table 3: Forecasting results (CRPS) on eight real-world datasets for various statistical and neural methods. Best scores in **bold**, second best underlined.

| Method | Electr. | Exchange | KDDCup | M4 (H) | Solar | Traffic | UberTLC | Wiki2000 |
|---|---|---|---|---|---|---|---|---|
| SN | 0.069±0.000 | 0.013±0.000 | 0.561±0.000 | 0.048±0.000 | 0.512±0.000 | 0.221±0.000 | 0.299±0.000 | 0.423±0.000 |
| ARIMA | 0.344±0.000 | 0.008±0.000 | 0.514±0.000 | 0.031±0.000 | 0.558±0.003 | 0.486±0.000 | 0.478±0.000 | 0.654±0.000 |
| ETS | 0.055±0.000 | 0.008±0.000 | 0.584±0.000 | 0.070±0.000 | 0.550±0.000 | 0.492±0.000 | 0.520±0.000 | 0.651±0.000 |
| DLinear | 0.058±0.001 | 0.015±0.004 | 0.318±0.015 | 0.055±0.007 | 0.794±0.027 | 0.131±0.000 | 0.250±0.006 | 0.259±0.002 |
| DeepAR | 0.051±0.000 | 0.013±0.004 | 0.362±0.017 | 0.045±0.013 | 0.429±0.055 | 0.103±0.002 | 0.168±0.002 | 0.215±0.003 |
| TFT | 0.060±0.001 | **0.007**±0.000 | 0.543±0.048 | 0.038±0.002 | 0.371±0.006 | 0.128±0.005 | 0.202±0.009 | 0.219±0.004 |
| WaveNet | 0.058±0.008 | 0.012±0.001 | 0.305±0.014 | 0.055±0.014 | 0.360±0.009 | 0.099±0.004 | 0.180±0.013 | **0.207**±0.003 |
| PatchTST | 0.055±0.000 | 0.010±0.001 | 0.420±0.011 | 0.034±0.004 | 0.728±0.015 | 0.151±0.007 | 0.219±0.004 | 0.209±0.001 |
| CSDI | 0.051±0.000 | 0.013±0.001 | 0.309±0.006 | 0.043±0.004 | 0.360±0.006 | 0.152±0.001 | 0.213±0.007 | 0.318±0.012 |
| SSSD | 0.048±0.001 | 0.010±0.001 | **0.274**±0.009 | 0.050±0.007 | 0.384±0.023 | 0.097±0.002 | 0.156±0.007 | 0.209±0.004 |
| Biloš et al. (2023) | 0.067±0.002 | 0.012±0.004 | 1.147±0.300 | - | 0.379±0.009 | 0.317±0.053 | 0.450±0.086 | 0.318±0.022 |
| TSDiff | 0.049±0.000 | 0.011±0.001 | 0.311±0.026 | 0.036±0.001 | 0.358±0.020 | 0.098±0.002 | 0.172±0.005 | 0.221±0.001 |
| TSFlow-Cond. (ISO) | **0.045**±0.001 | 0.008±0.000 | 0.303±0.004 | 0.034±0.001 | 0.350±0.016 | **0.082**±0.000 | 0.154±0.002 | 0.208±0.000 |
| TSFlow-Cond. (OU) | **0.045**±0.000 | 0.008±0.000 | 0.288±0.004 | 0.028±0.008 | 0.344±0.006 | **0.082**±0.000 | 0.154±0.002 | **0.207**±0.001 |
| TSFlow-Cond. (SE) | **0.045**±0.001 | 0.008±0.000 | 0.296±0.002 | **0.027**±0.006 | 0.343±0.001 | 0.083±0.000 | 0.155±0.002 | 0.211±0.003 |
| TSFlow-Cond. (PE) | **0.045**±0.001 | 0.008±0.000 | 0.278±0.002 | 0.033±0.005 | **0.339**±0.005 | 0.082±0.000 | 0.155±0.003 | 0.211±0.001 |
| TSFlow-Uncond. | 0.049±0.001 | 0.011±0.001 | 0.287±0.005 | 0.032±0.001 | 0.396±0.005 | 0.086±0.000 | **0.154**±0.002 | 0.291±0.005 |

We evaluate two variants of TSFlow: a conditional model that uses domain-specific Gaussian process regression prior (TSFlow-Cond., see Sec. 3.2), and an unconditional model that applies conditional prior sampling and guidance (TSFlow-Uncond., see Sec. 3.1.2 and 3.1.3). Tab. 3 shows that TS-Flow is competitive with statistical and neural baselines and achieves state-of-the-art results on various datasets. On 6/8 datasets, TSFlow attains the best scores with up to 14% improvement to the second-best method. Furthermore, TSFlow-Cond. outperforms or matches the diffusion-based baselines CSDI, SSSD, (Biloš et al., 2023), and TSDiff on 7/8 datasets while requiring fewer NFEs. Finally, we observe that, while TSFlow-Uncond. is competitive with other baselines, it is slightly inferior to the conditional versions of TSFlow, showing that the conditional training is an effective method to specialize TSFlow for forecasting. Furthermore, while all kernels yield similar results, the Ornstein-Uhlenbeck prior slightly improves over the others. We show example forecasts of TSFlow-Cond. on the Traffic dataset in Fig. 2 and for Electricity and Solar in App. A.9. Furthermore, we provide an ablation study of TSFlow's different components in App. B.2 and extend the forecasting setting in Apps. B.6 to B.8.

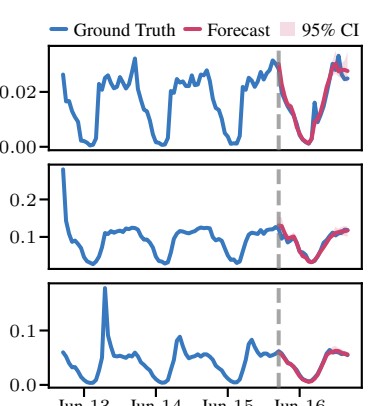

Figure 2: Example forecasts and ground truth of TSFlow-Cond. (OU) of the first three time series in the test set of Traffic.

## 5 RELATED WORK

**Diffusion Models and Conditional Flow Matching.** Diffusion models have been applied to several fields and achieved state-of-the-art performance (Ho et al., 2020; Hoogeboom et al., 2022; Lienen et al., 2024). Various works have demonstrated the effectiveness of unconditional models in conditional tasks through diffusion guidance (Epstein et al., 2023; Dhariwal & Nichol, 2021; Bansal et al., 2023; Nichol et al., 2021; Avrahami et al., 2022), solving inverse problems (Kawar et al., 2022), or iterative resampling (Lugmayr et al., 2022). Conditional Flow Matching (Lipman et al., 2022) is a recent approach proposed as an alternative to and generalization of diffusion models. This framework has been extended to incorporate couplings between data and prior samples, and trained to solve the dynamic optimal transport problem (Tong et al., 2023). Albergo et al. (2023) demonstrated the applicability of data-dependent couplings for stochastic interpolants to improve in-painting and

super-resolution image generation. Lastly, recent work shows the potential for conditional generation by solving an optimization problem and differentiating through the flow (Ben-Hamu et al., 2024).

**Generative Models for Time Series.** Various generative models have been adapted to time series modeling, including Generative Adversarial Networks (GANs) (Yoon et al., 2019), normalizing flows (Rasul et al., 2020; Alaa et al., 2020), and Variational Autoencoders (VAEs) (Desai et al., 2021). Recent works have successfully applied diffusion models to time series. The first approach, TimeGrad (Rasul et al., 2021), applies a diffusion model on top of an LSTM to perform autoregressive multivariate time series forecasting, which was later extended by Biloš et al. (2023) to settings with continuous time and non-isotropic noise distributions. Note that unlike TSFlow, Biloš et al. (2023) neither use optimal transport paths nor couplings. CSDI (Tashiro et al., 2021) and SSSD (Alcaraz & Strodthoff, 2022) perform forecasting and imputation via conditional diffusion models. While CSDI uses transformer layers, SSSD makes use of S4 layers (Gu et al., 2021) to model temporal dynamics. Lastly, TimeDiff (Shen & Kwok, 2023) explores conditioning mechanisms, and TSDiff (Kollovieh et al., 2023) introduced an unconditional diffusion model conditioned through diffusion guidance similar to our unconditional model.

Additionally, Tamir et al. (2024) model dynamical systems using conditional flow matching (CFM), where the time series and ODE share the same time dimension. In contrast, TSFlow operates on two orthogonal time dimensions, one for the generative process and another within the time series. Moreover, Kerrigan et al. (2023) extend flow-matching to infinite-dimensional spaces using Gaussian processes, but their focus differs from ours. While they construct conditional Gaussian measure paths, TSFlow directly parameterizes the joint data and prior distribution.

# 6 CONCLUSION

In this work, we introduced TSFlow, a novel conditional flow matching model for probabilistic time series forecasting. TSFlow leverages flexible, data-dependent prior distributions and optimal transport paths to enhance unconditional and conditional generative capabilities. Our experiments on eight real-world datasets demonstrated that TSFlow consistently achieves state-of-the-art-performance.

We found that non-isotropic Gaussian process priors, particularly the periodic kernel, often led to better performance than isotropic priors, even with fewer neural function evaluations (NFEs). The conditional version of TSFlow with Gaussian process regression priors showed improvements over diffusion-based approaches, further emphasizing the effectiveness of our proposed methods. Additionally, we demonstrated that the unconditional model can be effectively used in a conditional setting via Langevin dynamics, offering additional flexibility in various forecasting scenarios.

**Limitations and Future Work.** While TSFlow achieves state-of-the-art performance across multiple benchmarks and reduces the computational costs compared to its diffusion-based predecessors, it has primarily been tested on univariate time series. Furthermore, different datasets and tasks, e.g., unconditional and conditional generation, require kernel selection, which should be optimized on the validation set. Future work could extend our approach to multivariate time series by incorporating multivariate Gaussian processes that capture dependencies across dimensions. Additionally, TSFlow allows for arbitrary source distributions, enabling the use of more involved priors, such as those based on data statistics (Park et al., 2024) or neural forecasting methods with intractable likelihoods. Exploring such distributions could further enhance performance.

## REPRODUCIBILITY STATEMENT

To ensure the reproducibility of our results, we provide a detailed description of our experimental setup, including the benchmark datasets and evaluation metrics, in Sec. 4 and App. A.1. Additionally, we outline the hyperparameters used in App. A.2 and include the pseudocodes for our unconditional and conditional model in App. A.10.

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

## A  EXPERIMENTAL SETUP AND HYPERPARAMETERS

### A.1  DATASETS

We used eight commonly used datasets and train/test splits from GluonTS (Alexandrov et al., 2020) encompassing various domains and frequencies. In Tab. 4, we show an overview of the datasets.

Table 4: Overview of the datasets and their statistics used in our experiments.

| Dataset | Train Size | Test Size | Domain | Freq. | Median Seq. Length | Prediction Length |
|---|---|---|---|---|---|---|
| Electricity[a] | 370 | 2590 | $\mathbb{R}^+$ | H | 5833 | 24 |
| Exchange[b] | 8 | 40 | $\mathbb{R}^+$ | D | 6071 | 30 |
| KDDCup[c] | 270 | 270 | $\mathbb{N}$ | H | 10850 | 48 |
| M4 (H)[d] | 414 | 414 | $\mathbb{N}$ | H | 960 | 48 |
| Solar[e] | 137 | 959 | $\mathbb{R}^+$ | H | 7009 | 24 |
| Traffic[f] | 963 | 6741 | $(0, 1)$ | H | 4001 | 24 |
| UberTLC[g] | 262 | 262 | $\mathbb{N}$ | H | 4320 | 24 |
| Wikipedia[h] | 2000 | 10000 | $\mathbb{N}$ | D | 792 | 30 |

[a]https://archive.ics.uci.edu/ml/datasets/ElectricityLoadDiagrams20112014
[b]https://github.com/laiguokun/multivariate-time-series-data
[c]https://zenodo.org/record/4656756
[d]https://github.com/Mcompetitions/M4-methods/tree/master/Dataset
[e]https://www.nrel.gov/grid/solar-power-data.html
[f]https://zenodo.org/record/4656132
[g]https://github.com/fivethirtyeight/uber-tlc-foil-response
[h]https://github.com/mbohlkeschneider/gluon-ts/tree/mv_release/datasets

### A.2  HYPERPARAMETERS AND TRAINING DETAILS

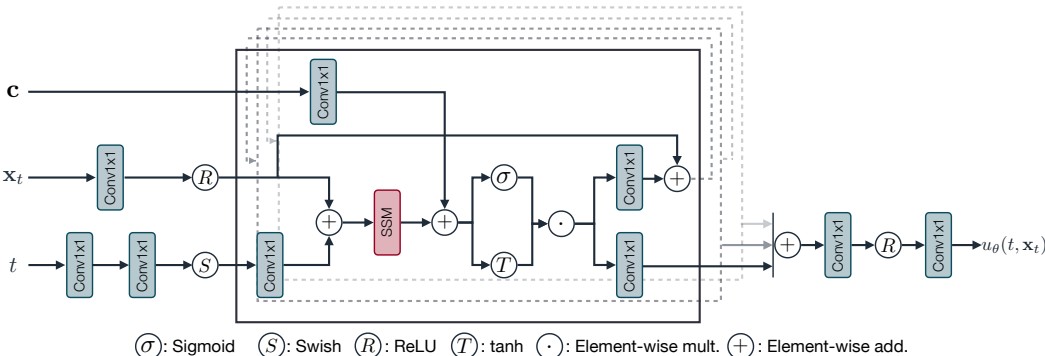

$\sigma$: Sigmoid    $S$: Swish    $R$: ReLU    $T$: tanh    $\cdot$: Element-wise mult.    $+$: Element-wise add.

Figure 3: Architecture of TSFlow. The architecture consists of three residual blocks containing an S4 layer operating across the time dimension. The input to the model are noisy time series $\mathbf{x}_t$, the timestep $t$, and in the case of the conditional setup, a condition $\mathbf{c}$, which contains $\mathbf{y}^p$ and a binary observation mask. The prediction is the approximated flow field $u_{\boldsymbol{\theta}}(t, \mathbf{x}_t)$.

We trained both the conditional and unconditional variants of TSFlow using the Adam optimizer with a learning rate of $10^{-3}$, applying gradient clipping with a threshold of 0.5. The conditional model was trained for 400 epochs, while the unconditional model was trained for 1000 epochs. Each epoch consists of 128 batches, with each batch containing 64 sequences. To stabilize the training process, we employed an Exponential Moving Average (EMA) of the model parameters with a decay rate of $\beta = 0.9999$. To parametrize the Gaussian processes, we choose $\ell = \sqrt{1/2}$, $\ell = 1$, and $\ell = \sqrt{2}$, for the SE, OU, and PE kernels, respectively, simplifying these even further.

TSFlow uses a context length equal to the prediction length and includes additional lag features in the feature vector $\mathbf{c}$. Since TSFlow-Uncond. is an unconditional model and does not utilize feature inputs, we increase the context window to 336 for hourly data and 210 for daily data.

The architecture of TSFlow is shown in Fig. 3 and consists of three residual blocks with a hidden dimension of 64, resulting in approximately 176,000 trainable parameters. We encode timesteps using 64-dimensional sinusoidal embeddings (Vaswani et al., 2017).

To solve the ordinary differential equation (ODE), we utilized an Euler solver with 32 steps. The unconditional version, TSFlow-Uncond., uses 16 additional NFEs due to the additional conditional prior sampling, which involves 4 iterations with 4 steps each. The unconditional version in Tab. 2 uses 4 steps. TSFlow-Uncond. performs four iterations to sample from the prior distribution with a step size of $\eta = 0.005$ and a noise scale of $0.5$. The guidance scale parameter is selected between 8 and 32 based on the validation set's performance. Finally, we choose $\kappa$ in the guidance score uniformly between $0.1$ and $0.9$. An overview of the hyperparameters and training details is provided in Table 5.

Table 5: Hyperparameters of TSFlow.

| Hyperparameter | Value |
|---|---|
| Learning rate | $10^{-3}$ |
| Optimizer | Adam |
| Batch size | 64 |
| Gradient clipping threshold | 0.5 |
| Epochs | 400 / 1000 |
| EMA momentum | 0.9999 |
| Residual blocks | 3 |
| Residual channels | 64 |
| Time Emb. Dim. | 64 |
| ODE Steps | 32 / 16 / 4 |
| ODE Solver | Euler |
| Guidance strength $s$ | 8 / 16 / 24 / 32 |
| Quantile $\kappa$ | $\mathcal{U}[0.1, 0.9]$ |
| $\sigma_{\min}$ | $10^{-4}$ |
| $\sigma_{\max}$ | $10^{-4}$ / 1 |

**Normalization.** To account for varying magnitudes across time series within the dataset, we normalize each series by dividing its values by the global mean of the training set:

$$\mathbf{x}_p^{\text{norm}} = \frac{\mathbf{x}_p}{\text{mean}(\mathbf{x}_{\text{hist}})},$$

where $\text{mean}$ denotes the mean aggregation function, and $\mathbf{x}_{\text{hist}}$ encompasses the entire (train) time series beyond the context window.

### A.3 ASYMMETRIC LAPLACE DISTRIBUTION

In Sec. 3.1.2 and 3.1.3, we parameterize the guidance score using an Asymmetric Laplace Distribution (ALD). The probability density function (PDF) is defined with a location parameter $m$, a scale parameter $\lambda$, and a quantile $\kappa$:

$$p(x; m, \lambda, \kappa) \propto \exp\left(-\frac{1}{\lambda}\max\{\kappa \cdot (x - m), (\kappa - 1) \cdot (x - m)\}\right). \tag{15}$$

By setting $\lambda = 1$, the guidance score from Eq. (12) simplifies to the quantile loss:

$$\nabla_{\mathbf{x}_t} \log p_t(\mathbf{y}^p \mid \mathbf{x}_t) = \max\{\kappa \cdot (\mathbf{y}^p - \phi_{\boldsymbol{\theta},1}(\mathbf{x}_t)), (\kappa - 1) \cdot (\mathbf{y}^p - \phi_{\boldsymbol{\theta},1}(\mathbf{x}_t))\}, \tag{16}$$

which corresponds to a weighted $\ell_1$ loss. In practice, we select $\kappa$ uniformly from the range $(0.1, 0.9)$ to ensure coverage across different quantiles.

A.4 BASELINES

We utilize the official implementations provided by the respective authors for CSDI, TSDiff, and Biloš et al. (2023). Although the original CSDI paper focuses on multivariate time series experiments, their codebase supports univariate forecasting, allowing us to use it without any modifications with their recommended hyperparameters and training setup. As SSSD targeted primarily multivariate datasets, we adjusted their model to the univariate case. Specifically, to avoid overfitting, we set the hidden dimension to 64 and the number of residual layers to 3, aligning the network sizes with those of TSDiff and TSFlow. For PatchTST, ARIMA, and ETS, we use AutoGluon (Shchur et al., 2023). For the remaining baselines, we employ the publicly available implementations in GluonTS (Alexandrov et al., 2020) with their recommended hyperparameters.

A.5 METRICS

**Linear Predictive Score (LPS)**    To assess the performance of the unconditional models in downstream forecasting, we use the Linear Predictive Score (LPS) as proposed by Kollovieh et al. (2023). The LPS is computed as the test CRPS of a linear ridge regression model trained on synthetic samples generated by the model. Specifically, the linear model is trained to map the observed past of a time series, $\mathbf{y}^p \in \mathbb{R}^{L^p}$, to its future values $\mathbf{y}^f \in \mathbb{R}^{L^f}$. The linear model is regressed against the true future values, using the past $\mathbf{y}^p$ as input, allowing us to measure how well the synthetic samples capture the relationship between past and future. Since the linear ridge regression can be fit in closed form, it remains robust to random initialization. Following Kollovieh et al. (2023), we generate 10,000 synthetic samples to fit the linear model.

A.6 UNCONDITIONAL PRIORS

Since the observations occur at discrete time steps, i.e., $\tau_i = i$ for $i = 0, \ldots, L - 1$, we normalize them to align with the periodicity of the dataset. More specifically, we adjust the time steps using the formula:

$$\tau_i^{\text{norm}} = \frac{\tau_i \cdot \pi}{p}, \tag{17}$$

where $p$ represents the frequency length of the time series; for instance, 24 for hourly data and 30 for daily data. This normalization scales the time steps so that one full period corresponds to $\pi$, ensuring that the periodic kernel accurately captures the cyclical patterns in the data. Examples of the different processes for various kernels are illustrated in Fig. 4. To avoid numerical issues such as ill-conditioned covariance matrices, we extend each kernel $K$ with a white noise kernel $K_{\text{white}}(\tau, \tau') = \delta(\tau - \tau')$ to $K'(\tau, \tau') = K(\tau, \tau') + K_{\text{white}}(\tau, \tau')$. Adding this white noise ensures that the covariance matrix is positive definite and improves numerical stability during computations.

A.7 CONDITIONAL PRIORS

Before applying Gaussian process regression, we apply a frequency-based z-score normalization to the data, which standardizes each segment according to its periodicity. For each data point $\mathbf{x}_i$, the transformation is defined as:

$$\mathbf{x}_i^{\text{norm}} = \mathbf{x}_i - \mu_p, \tag{18}$$

where $\mu_p$ is the mean computed over segments of length corresponding to the periodicity $p$ of the time series. This normalization ensures that each segment of the data, organized by its periodic frequency, has a mean of zero.

A.8 RUNTIME

In App. A.8, we provide the runtime of TSFlow on the Solar dataset for both training and evaluation on an Nvidia A100 GPU. The evaluation time includes generating 100 sample forecasts for each time series in the dataset, i.e., generating 95900 samples.

We observed minimal differences in runtime between the isotropic and Gaussian process regression (GPR) priors. We attribute this minor difference to the computations in Eq. (14). Apart from a single

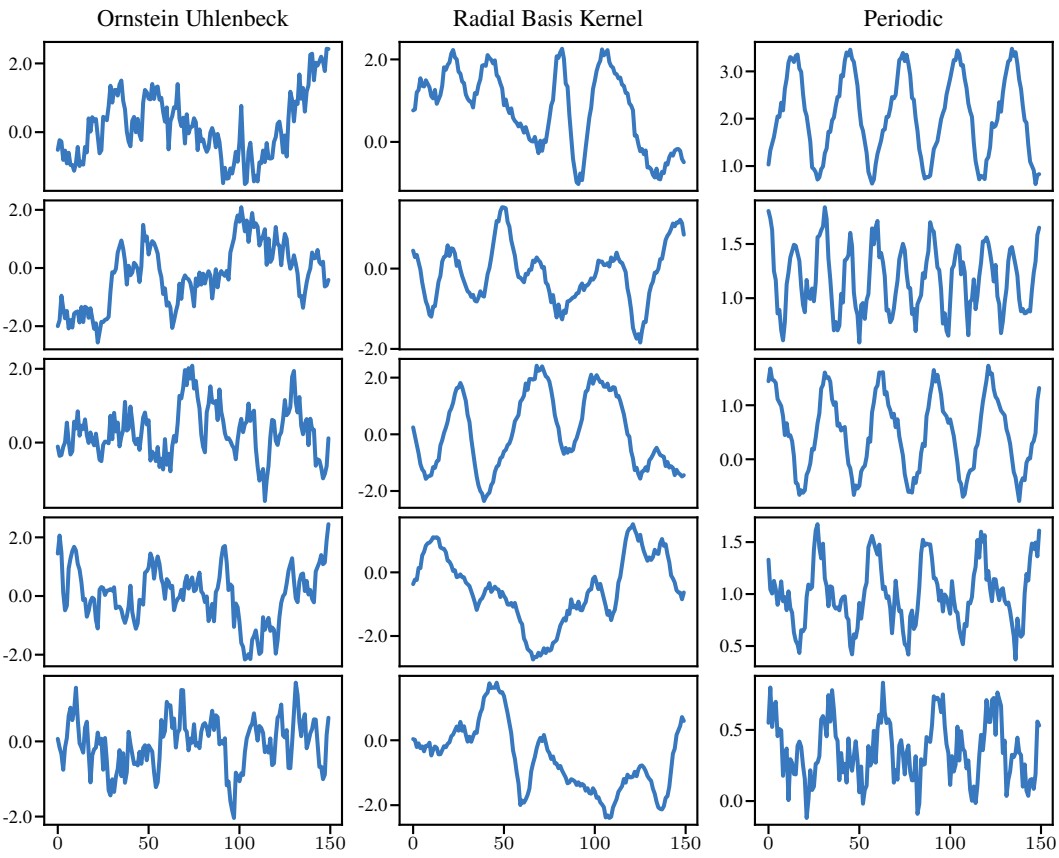

Figure 4: Examples of samples from the unconditional Gaussian processes for $L = 150$ and $p = 30$.

Table 6: Comparison of different configurations with their corresponding training and evaluation times.

| Configuration | Train - Epoch | Eval. |
|---|---|---|
| TSFlow-Cond. – Isotropic prior | 2.94 | 24.29 |
| TSFlow-Cond. – GPR prior | 3.17 | 24.65 |
| TSFlow-Uncond. – Random | 3.66 | – |
| TSFlow-Uncond. – Optimal Transport | 3.77 | – |
| TSFlow-Uncond. – CPS + Guidance | – | 81.97 |

matrix-vector multiplication, all computations only need to be performed once, which minimizes the overhead associated with the GPR prior.

In the unconditional model, we observed a neglectable computational overhead when using optimal transport maps during training. For evaluation, however, it takes approximately three times longer than the standard conditional version. We attribute this increase to two factors: the use of a longer context window (see App. A.2) and the need to differentiate through the ODE integration process.

## A.9  ADDITIONAL FORECASTS

We show more example forecasts of TSFlow-Cond. on Solar and Electricity in Fig. 5

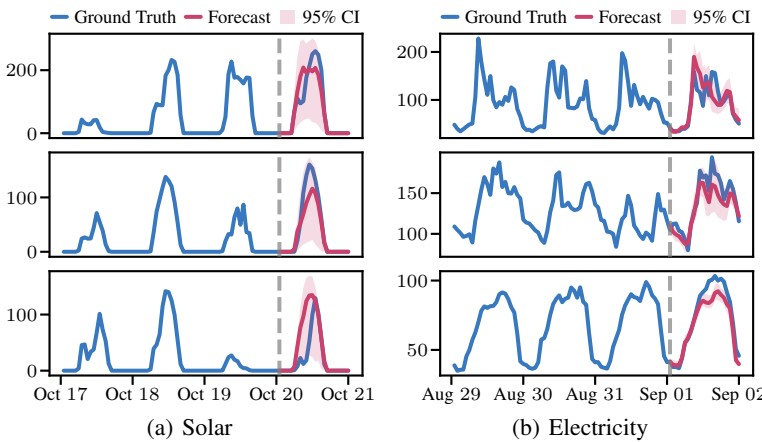

Figure 5: Example forecasts of TSFlow-Cond. on the test set of the two datasets Solar and Electricity.

## A.10 ALGORITHMS

### A.10.1 TRAINING ALGORITHMS FOR TSFLOW

We provide the training algorithm for the unconditional and conditional versions in Alg. 1 and Alg. 2.

---

**Algorithm 2: Conditional Training of TSFlow**

---

1: **Input:** Conditional prior distribution $q_0$, data distribution $q_1$, noise level $\sigma_t$, network $u_{\boldsymbol{\theta}}$
2: **for** iteration $= 1, \dots$ **do**
3:   $\mathbf{y} \sim q_1(\mathbf{y}); \; \mathbf{x}_0 \sim q_0(\mathbf{x}_0 \mid \mathbf{y}^p)$    $\triangleright$ sample batches from the conditional prior and dataset
4:   $t \sim \mathcal{U}(0,1)$                $\triangleright$ sample random time step $t$
5:   $\boldsymbol{\mu}_t \leftarrow t\mathbf{y} + (1-t)\mathbf{x}_0$           $\triangleright$ compute mean of $p_t$
6:   $\mathbf{x}_t \sim \mathcal{N}(\boldsymbol{\mu}_t, \sigma_t^2 \mathbf{I})$           $\triangleright$ sample from $p_t(\cdot \mid \mathbf{z})$
7:   $\mathcal{L}(\boldsymbol{\theta}) \leftarrow \|u_{\boldsymbol{\theta}}(t, \mathbf{x}_t, \mathbf{y}^p) - u_t(\mathbf{x}_t \mid \mathbf{z})\|^2$    $\triangleright$ regress conditional vector field
8:   $\boldsymbol{\theta} \leftarrow \text{Update}(\boldsymbol{\theta}, \nabla_{\boldsymbol{\theta}} \mathcal{L}(\boldsymbol{\theta}))$        $\triangleright$ gradient step
9: **end for**
10: **Return:** $u_{\boldsymbol{\theta}}$

---

## A.11 EVALUATION ALGORITHMS FOR TSFLOW

We provide a formal description of the conditional prior sampling algorithm in Alg. 3.

---

**Algorithm 3: Conditional Prior Sampling (without generation)**

---

1: **Input:** Prior distribution $q_0$, network $u_{\boldsymbol{\theta}}$, observation $\mathbf{y}^p$, prior sample $\mathbf{x}_0^{(0)}$
2: **for** iteration $= 1, \dots$ **do**
3:   $\xi_i \sim \mathcal{N}(\mathbf{0}, \mathbf{I})$              $\triangleright$ sample noise
4:   $\mathbf{x}_0^{(i+1)} = \mathbf{x}_0^{(i)} - \eta \nabla_{\mathbf{x}_0} \log q_0(\mathbf{x}_0^{(i)} \mid \mathbf{y}^p) + \sqrt{2\eta}\xi_i$    $\triangleright$ update $\mathbf{x}_0$
5: **end for**
6: **Return:** $\mathbf{x}_0^{(i+1)}$

---

# B ADDITIONAL RESULTS

## B.1 EFFECT ON THE OPTIMAL TRANSPORT PROBLEM.

To understand the effect of the prior distribution, we measure the Wasserstein distance between batches $\{\mathbf{x}_0^{(i)} \sim q_0\}$ drawn from the prior and batches $\{\mathbf{y}^{(i)} \sim q_1\}$ drawn from the data distribution across four common benchmark datasets in time series forecasting. Our results in Fig. 6 show

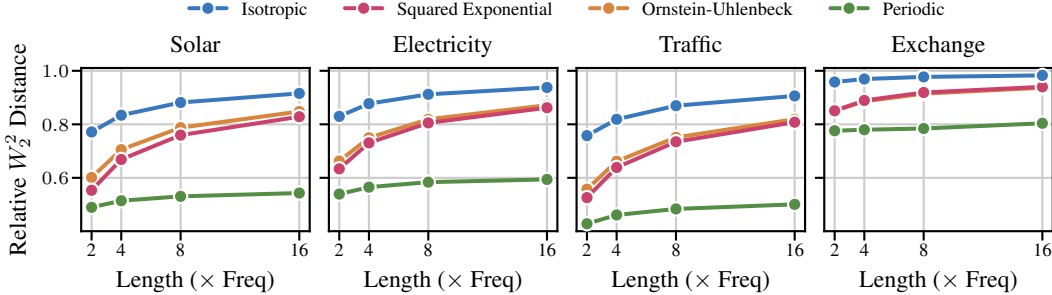

Figure 6: Mini-batch Wasserstein $W_2^2$ distances between samples from prior and data on common benchmark datasets. x-axis shows data dimension in multiples of each dataset's base period, e.g., 24h for Solar. Distances relative to the transport distance from a random transport map as used in CFM.

that Gaussian process priors reduce the Wasserstein distance between the prior and data distribution if the kernel exploits characteristics of the data. When we parametrize the prior with SE and OU kernels, the Wasserstein distance to the data distribution decreases substantially, although it slowly converges to an isotropic prior as the sequence length increases. Notably, the periodic kernel achieves the largest reduction in distance on these datasets with only marginal growth as the sequence length increases because it effectively captures the periodicity — a common feature of real-world time series — especially well. These findings support our hypothesis that an informed choice of prior distribution can substantially affect the learning problem.

## B.2 ABLATION

In the following, we describe and ablate different components of our conditional model. We use the same experimental setup as described in App. A.2. The base model corresponds to a basic Conditional Flow Matching model with an isotropic prior distribution.

**Ornstein Uhlenbeck Prior.** The CFM framework allows one to accommodate arbitrary prior distributions. Instead of using an isotropic Gaussian, we use a Gaussian Process regression based on the Ornstein Uhlenbeck kernel as explained in Sec. 3.2. The results demonstrate an improvement of the CRPS on all datasets, demonstrating the advantage of conditional prior distributions.

**Lags.** To extend information into the past without increasing the context window, we use lagged values as features. More specifically, we use the default lags deployed by GluonTS (Alexandrov et al., 2020), which are also used by various baselines.

**Bidirectional S4 layers.** To allow information flow in both temporal directions, we can employ bidirectional S4 layers. This allows the model to generate consistent forecasts and slightly improves the results.

**Exponential Moving Average.** To stabilize training, we employ exponential moving average (EMA), which keeps a copy $\boldsymbol{\theta}'$ of the parameters $\boldsymbol{\theta}$ and is updated via a momentum update after each gradient step:

$$\boldsymbol{\theta}' \leftarrow m\boldsymbol{\theta}' + (1 - m)\boldsymbol{\theta},$$

where $m$ is a momentum parameter. To generate forecasts, we use the conditional vector field $u_{\boldsymbol{\theta}'}$ instead of $u_{\boldsymbol{\theta}}$.

Table 7: Ablation forecasting results (CRPS) for TSFlow-Cond. on five real-world datasets. Best scores in **bold**, second best underlined.

| Method | Electr. | KDDCup | Solar | Traffic | UberTLC |
|---|---|---|---|---|---|
| Basemodel | 0.058±0.002 | 0.302±0.010 | 0.339±0.008 | 0.133±0.001 | 0.199±0.004 |
| + OU Prior | 0.049±0.000 | **0.273**±0.007 | **0.333**±0.006 | 0.107±0.001 | 0.161±0.004 |
| + Lags | 0.046±0.001 | 0.392±0.025 | 0.356±0.017 | 0.084±0.001 | 0.154±0.002 |
| + Bidir. | 0.046±0.001 | 0.285±0.010 | 0.341±0.006 | **0.083**±0.000 | **0.153**±0.005 |
| + EMA | **0.045**±0.001 | **0.273**±0.004 | 0.343±0.002 | **0.083**±0.000 | **0.153**±0.001 |

## B.3 CONDITIONAL PRIOR SAMPLING

We present the effect of conditional prior sampling (CPS) in Table 8 on the benchmark datasets, comparing it to a model that employs only guidance without CPS.

Table 8: Effect of the CPS iterations on the forecasting results (CRPS) for TSFlow-Uncond. on eight real-world datasets. Best scores in **bold**.

| #It. | Electr. | Exchange | KDDCup | M4Hourly | Solar | Traffic | UberTLC | Wiki2000 |
|---|---|---|---|---|---|---|---|---|
| 0 | **0.049**±0.001 | 0.012±0.001 | **0.294**±0.002 | **0.029**±0.001 | **0.371**±0.006 | 0.096±0.000 | 0.154±0.001 | 0.290±0.004 |
| 4 | **0.049**±0.001 | **0.011**±0.001 | 0.299±0.004 | **0.029**±0.001 | 0.464±0.004 | **0.089**±0.000 | **0.153**±0.002 | **0.279**±0.007 |

As shown in the table, on 7 out of 8 datasets, incorporating CPS yields equal or better CRPS scores compared to the model that uses only guidance. The exception is the Solar dataset, where CPS performs worse than the guided-only model. This suggests that solely using guidance is insufficient for effectively conditioning TSFlow-Uncond. during inference.

## B.4 GUIDANCE SCORE FUNCTION

In Eqs. (9) and (12), we utilize an asymmetric Laplace distribution (ALD) to model the guidance score. Alternatively, this score could be modeled using a Gaussian distribution, which simplifies to a mean squared error loss after applying the logarithm. As demonstrated in Tab. 9, the asymmetric

Table 9: Effect of the score function on the forecasting results (CRPS) for TSFlow-Uncond. on eight real-world datasets. Best scores in **bold**.

| Distr. | Electr. | Exchange | KDDCup | M4Hourly | Solar | Traffic | UberTLC | Wiki2000 |
|---|---|---|---|---|---|---|---|---|
| Gaussian | 0.059±0.001 | 0.020±0.005 | 0.317±0.016 | 0.031±0.000 | **0.370**±0.010 | 0.101±0.001 | 0.169±0.005 | 24.046±6.368 |
| ALD | **0.049**±0.001 | **0.011**±0.001 | **0.299**±0.004 | **0.029**±0.001 | 0.464±0.004 | **0.089**±0.000 | **0.153**±0.002 | **0.279**±0.007 |

Laplace distribution yields better results across 6 out of 8 datasets. We attribute this to the fact that different quantile parameters $\kappa$ take the whole distribution into account rather than focusing on a single point estimate.

## B.5 NON-GAUSSIAN PRIOR DISTRIBUTIONS

In addition to Gaussian process priors, TSFlow support non-Gaussian priors for $q_0(\mathbf{x}_0^f \mid \mathbf{y}^p)$. Instead of parameterizing the prior via Gaussian process regression, we define it as

$$q_0(\mathbf{x}_0^f \mid \mathbf{y}^p) = \mathcal{N}\left(\mathbf{x}_0^f \mid g(\mathbf{y}^p), \mathbf{I}\right),$$

where $g$ is a base forecaster. We demonstrate this approach using the Seasonal Naive (SN) forecaster and compare it to the Ornstein-Uhlenbeck prior in Tab. 10.

## B.6 NEURAL FUNCTION EVALUATIONS

In Tab. 1, we present the Wasserstein-2 distances for 4 and 16 Neural Function Evaluations (NFEs). To further illustrate the performance, we show Fig. 7 the Wasserstein-2 distances across varying NFEs for selected datasets. As demonstrated in the figure, the non-isotropic priors achieve better performance than the isotropic prior.

Table 10: Forecasting performance (CRPS) on eight datasets for the seasonal naive (SN) prior compared to the Ornstein-Uhlenbeck prior. Best scores are in **bold**.

| Prior | Electr. | Exchange | KDDCup | M4 (H) | Solar | Traffic | UberTLC | Wiki |
|---|---|---|---|---|---|---|---|---|
| SN | **0.044** ±0.001 | **0.008** ±0.000 | 0.321 ±0.010 | 0.036 ±0.001 | 0.348 ±0.006 | **0.082** ±0.000 | 0.154 ±0.002 | 0.233 ±0.002 |
| OU | 0.045 ±0.001 | 0.009 ±0.001 | **0.273** ±0.004 | **0.025** ±0.001 | **0.343** ±0.002 | 0.083 ±0.000 | **0.153** ±0.001 | **0.227** ±0.000 |

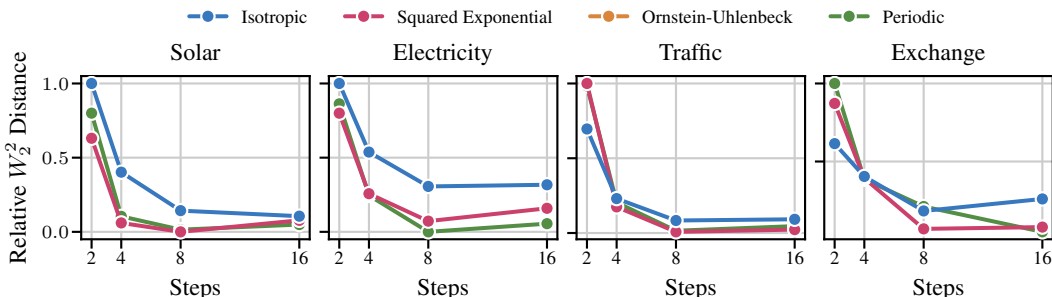

Figure 7: Wasserstein $W_2^2$ distances between samples from unconditional generative model and data on common benchmark datasets. x-axis shows NFEs using an Euler solver.

## B.7 FORECASTING WITH MISSING VALUES

In our experiments, we focus on forecasting due to the availability of standardized benchmarks and to ensure fair comparisons with autoregressive models, which do not benefit from bidirectional information passing. However, TSFlow is not restricted to a forecasting setting and can be extended to arbitrary conditional generation tasks by adjusting the observation mask.

To demonstrate this, we evaluate forecasting scenarios with missing values in the context window. Following the setup by Kollovieh et al. (2023), we test three missing values scenarios with 50% missing rate: (1) random missing (RM), where observed values are randomly masked; (2) blackout missing beginning (BO-B), where values are missing at the start of the context window; and (3) blackout missing end (BO-E), where values are missing at the end of the context window just before the forecast. For these experiments, we use the conditional version of TSFlow with the OU kernel and compare it against TSDiff, reporting the best result (conditional or unconditional) for TSDiff in each case. The results for the three setups are presented in Tabs. 11 to 13, respectively.

Table 11: Forecasting results (CRPS) with missing values in the context window (RM). Best scores in **bold**.

| Method | Electricity | Exchange | KDDCup | Solar | Traffic | UberTLC |
|---|---|---|---|---|---|---|
| TSFlow | **0.048**±0.001 | **0.009**±0.001 | 0.432±0.121 | 0.362±0.006 | **0.085**±0.000 | **0.157**±0.002 |
| TSDiff | 0.052±0.001 | 0.012±0.004 | **0.397**±0.042 | **0.357**±0.023 | 0.097±0.003 | 0.180±0.015 |

Overall, TSFlow outperforms both versions of TSDiff in 12 out of 18 cases, demonstrating its robustness across these scenarios.

## B.8 MULTIVARIATE FORECASTING

We extend TSFlow to the multivariate setting by incorporating a transformer across the feature dimension, following the approach of CSDI (Tashiro et al., 2021). To evaluate its effectiveness, we compare this multivariate extension (MV) to a channel-independent variant (CIDP) in Table 14. For reference, we also report the forecasting performance (CRPS-Sum) of CSDI (Tashiro et al., 2021) and the method proposed by Biloš et al. (2023). The results show that yields the best results on 3 out of 4 datasets.

Table 12: Forecasting results (CRPS) with missing values in the context window (BO-B). Best scores in **bold**.

| Method | Electricity | Exchange | KDDCup | Solar | Traffic | UberTLC |
|--------|-------------|----------|--------|-------|---------|---------|
| TSFlow | **0.047**±0.001 | **0.008**±0.001 | **0.305**±0.013 | 0.396±0.026 | **0.083**±0.000 | **0.158**±0.008 |
| TSDiff | 0.049±0.001 | 0.009±0.000 | 0.441±0.096 | **0.377**±0.017 | 0.094±0.005 | 0.181±0.009 |

Table 13: Forecasting results (CRPS) with missing values in the context window (BO-E). Best scores in **bold**.

| Method | Electricity | Exchange | KDDCup | Solar | Traffic | UberTLC |
|--------|-------------|----------|--------|-------|---------|---------|
| TSFlow | **0.061**±0.001 | 0.056±0.004 | 0.384±0.033 | 0.418±0.014 | **0.106**±0.002 | **0.165**±0.002 |
| TSDiff | 0.065±0.003 | **0.020**±0.001 | **0.344**±0.012 | **0.376**±0.036 | 0.123±0.023 | 0.179±0.013 |

Table 14: Forecasting performance (CRPS-Sum) on four datasets. Best scores are in **bold**.

| Model | Electricity | Exchange | Solar | Traffic |
|-------|-------------|----------|-------|---------|
| Biloš et al. (2023) | 0.0260 ±0.0011 | 0.0071 ±0.0009 | 0.3701 ±0.0179 | 1.2679 ±1.0624 |
| CSDI | 0.0184 ±0.0008 | 0.0076 ±0.0022 | 0.3179 ±0.0197 | 0.0221 ±0.0012 |
| TSFlow (CIDP) | 0.0196 ±0.0006 | **0.0060** ±0.0004 | 0.3780 ±0.0074 | 0.0240 ±0.0005 |
| TSFlow (MV) | **0.0176** ±0.0012 | 0.0073 ±0.0005 | **0.2792** ±0.0154 | **0.0217** ±0.0013 |

## C  GUIDED GENERATION

To enable guided generation, we use slightly modified conditional probability paths defined as:

$$p_t(\mathbf{x}_t \mid \mathbf{z}) = \mathcal{N}\left(\mathbf{x}_t; \boldsymbol{\mu}_t, \sigma_t^2 \mathbf{I}\right), \tag{19}$$

where $\boldsymbol{\mu}_t = (1-t)\mathbf{x}_0 + t\mathbf{x}_1$, $\sigma_t = (1-t)\sigma_{\max} + t\sigma_{\min}$, and $\sigma_{\max}$ and $\sigma_{\min}$ are constant standard deviations with $\sigma_{\max} > \sigma_{\min}$. Note that when $\sigma_{\max} = \sigma_{\min}$, we recover the same probability path as in Sec. 2.2. However, we require $\sigma_{\max} > \sigma_{\min}$ to obtain a score function in our vector field.

Using Theorem 3 from Lipman et al. (2022), we construct the corresponding vector field:

$$u_t(\mathbf{x}_t \mid \mathbf{z}) = \mathbf{x}_1 - \mathbf{x}_0 + \frac{\sigma_{\min} - \sigma_{\max}}{(1-t)\sigma_{\max} + t\sigma_{\min}}\left(\mathbf{x}_t - \mu_t\right). \tag{20}$$

We observe that the vector field can be expressed using the score function of the conditional probability paths in Eq. (19):

$$u_t(\mathbf{x}_t \mid \mathbf{z}) = \mathbf{x}_1 - \mathbf{x}_0 - (\sigma_{\min} - \sigma_{\max})\sigma_t \nabla_{\mathbf{x}_t} \log p_t(\mathbf{x}_t \mid \mathbf{z}) \tag{21}$$

Here, we used the fact that the score function of the Gaussian in Eq. (19) is:

$$\nabla_{\mathbf{x}_t} \log p_t(\mathbf{x}_t \mid \mathbf{z}) = -\frac{\mathbf{x}_t - (1-t)\mathbf{x}_0 - t\mathbf{x}_1}{\left((1-t)\sigma_{\max} + t\sigma_{\min}\right)^2}. \tag{22}$$

**Marginal Vector Field.**  By integrating over $\mathbf{z}$ we can rewrite the marginal vector field:

$$u_t(\mathbf{x}_t) = \int \frac{u_t(\mathbf{x}_t \mid \mathbf{z})p_t(\mathbf{x}_t \mid \mathbf{z})q(\mathbf{z})}{p_t(\mathbf{x}_t)}\mathrm{d}\mathbf{z} \tag{23}$$

$$= \int \frac{(\mathbf{x}_1 - \mathbf{x}_0 - (\sigma_{\min} - \sigma_{\max})\sigma_t \nabla_{\mathbf{x}_t} \log p_t(\mathbf{x}_t \mid \mathbf{z}))p_t(\mathbf{x}_t \mid \mathbf{z})q(\mathbf{z})}{p_t(\mathbf{x}_t)}\mathrm{d}\mathbf{z} \tag{24}$$

$$= \int \frac{(\mathbf{x}_1 - \mathbf{x}_0)p_t(\mathbf{x}_t \mid \mathbf{z})q(\mathbf{z})}{p_t(\mathbf{x}_t)}\mathrm{d}\mathbf{z} \tag{25}$$

$$-(\sigma_{\min} - \sigma_{\max})\sigma_t \int \frac{\nabla_{\mathbf{x}_t} \log p_t(\mathbf{x}_t \mid \mathbf{z})p_t(\mathbf{x}_t \mid \mathbf{z})q(\mathbf{z})}{p_t(\mathbf{x}_t)}\mathrm{d}\mathbf{z} \tag{26}$$

$$= v_t(\mathbf{x}_t) - (\sigma_{\min} - \sigma_{\max})\sigma_t \nabla_{\mathbf{x}_t} \log p_t(\mathbf{x}_t), \tag{27}$$

where $v_t(\mathbf{x}_t)$ represents the expected drift term $\mathbb{E}_{\mathbf{z}\mid\mathbf{x}_t}\left[\mathbf{x}_1 - \mathbf{x}_0\right]$.

**Conditional Score Function.**  To incorporate the observed past $\mathbf{y}^p$, we substitute the score function with its conditional counterpart:

$$\nabla_{\mathbf{x}_t} \log p_t(\mathbf{x}_t \mid \mathbf{y}^p) = \nabla_{\mathbf{x}_t} \log p_t(\mathbf{x}_t) + \nabla_{\mathbf{x}_t} \log p_t(\mathbf{y}^p \mid \mathbf{x}_t), \tag{28}$$

leading to the following modified vector field:

$$\tilde{u}_t(\mathbf{x}_t) = \underbrace{v_t(\mathbf{x}_t) - (\sigma_{\min} - \sigma_{\max})\sigma_t \nabla_{\mathbf{x}_t} \log p_t(\mathbf{x}_t)}_{=u_t(x_t)} - s(\sigma_{\min} - \sigma_{\max})\sigma_t \nabla_{\mathbf{x}_t} \log p_t(\mathbf{y}^p \mid \mathbf{x}_t), \tag{29}$$

where $s$ is a scaling parameter introduced to control the strength of the conditioning. We can summarize this as:

$$\tilde{u}_t(\mathbf{x}_t) = u_t(\mathbf{x}_t) - s\sigma_t \nabla_{\mathbf{x}_t} \log p_t(\mathbf{y}^p \mid \mathbf{x}_t), \tag{30}$$

where we have observed $\sigma_{\min} - \sigma_{\max}$ into $s$. Thus, to sample from the conditional generative model $p_t(\mathbf{x}_t \mid \mathbf{y}^p)$, we use the modified vector field $\tilde{u}_t(\mathbf{x}_t)$ from Eq. (30) with our trained vector field $u_{\boldsymbol{\theta}}(t, \mathbf{x}_t)$. In our unconditional experiments we use $\sigma_{\min} = 10^{-4}$ and $\sigma_{\max} = 1$.

