# OpenReview forum: "Flow Matching with Gaussian Process Priors for Probabilistic Time Series Forecasting"
_ICLR.cc/2025/Conference — ICLR 2025 Poster_

### Official Review · Reviewer_k8UA · 2024-10-20

**Soundness:** 3
**Presentation:** 2
**Contribution:** 3
**Rating:** 5
**Confidence:** 3

**Summary:**

This paper proposes a model that uses flow-matching algorithms to generate and forecast univariate time-series. When trained unconditionally, it replaces the isotropic Gaussian prior distribution with an informed one and uses conditional sampling and inferencing. The model itself can also be made conditional. Experiments show that the model surpasses existing ones on benchmark tasks.

**Strengths:**

1. The model is flexible with many possible options; the informed prior sampling is intuitive.

2. The model shows promising results on univariate benchmark tasks.

**Weaknesses:**

1. The model right now is only discussed and validated on univariate time-series.

2. While there are many proposed options in the model, there lacks an ablation or a comprehensive comparison of different choices. Neither is there any theoretical insights in the proposed model.

3. The presentation of the model needs a bit more clarification. The problem formulation can potentially be expanded a bit more (see questions below). The discussion of different training/inference choices is a bit dense. Maybe some concise pseudocode or flowchart in the main text could be helpful.

**Questions:**

1. If my understanding is correct, the overall idea of the model is to push an initial GP to the target time-series. Therefore, the sequence length $L$ in the problem formulation section is equivalent to the vector field dimension $d$ in the background section. Can the author(s) kindly confirm if this is true? It would be nice to clarify the setting a little bit. I think the main source of confusion is that there are two time indices in this paper: the time in flow-matching and the time in the time-series, and these are orthogonal to each other. I think this should be made clear somewhere in the paper.

2. Following the first question, it would be nice to indicate in Figure 1 the time-series index and the flow-matching index. Moreover, different notations should be used. (For instance, on line 202-203, $t$ should not be used for the kernel because it has already been used in Eq. (1).)

3. How would the model respond to a growing $L$? That is, there are two subquestions in this query:
	a. What is the time complexity of the model and how does it compare to other models that you benchmarked against?
	b. When $L$ is large, Eq. (6) seems to suffer from the curse of dimensionality and the integral would be impossible to discretize. How does the model see this issue, or is it not relevant?

4. The section "Effect on the Optimal Transport Problem" only considers the distance between sequences but not anything about the training. Can you show some experiments where models that use the periodic kernel are indeed easier to train on tasks that involve periodicity?

5. On line 302-303, you wrote "we additionally condition the prior distribution on the observed past data by approximating $q_0(\mathbf{x}_0 | \mathbf{y}^p)$ with $q_0(\mathbf{x}_0 | \mathbf{y}^p)$." I assume there is a typo. What would be the intended sentence?

---

> ### Author Response · Authors · 2024-11-21
>
> We thank the reviewer for their comprehensive feedback. In the following, we address their concerns.
>
>
> **Comment:** The model is only discussed and validated on univariate time-series.
> **Response:** We have evaluated our model on univariate time series as they are present in many real-world problems, and, more importantly, metrics specifically designed for multivariate forecasting, such as the CRPS-sum, have several limitations, particularly when it comes to evaluating the performance of individual dimensions. As shown in [1], the CRPS-sum can obscure the performance of the model on each dimension due to its aggregation process.
>
> Technically, TSFlow is not limited to univariate time series and can be extended to the multivariate case by adjusting the neural network $u_\theta(t,x_t)$ to handle multiple features, i.e., $u_\theta(t,x_t):[0,1]\times\mathbb{R}^{K\times L}\to\mathbb{R}^{K\times L}$.
>
> For this, one could use a channel-independent architecture such as PatchTST [2] or employ a feature interaction layer (e.g., linear layer or a transformer) across the feature dimension to process the multivariate time series.
>
>
> **Comment:** While there are many proposed options in the model, there lacks an ablation or a comprehensive comparison of different choices. Neither is there any theoretical insights in the proposed model.
> **Response:** We have extended our initial ablation in Appendix B.1. We ablate different components of our conditional model, number of neural function evaluations, compare our unconditional model with and without conditional prior sampling, and finally, we compare the asymmetric Laplace distribution against a Gaussian as guidance score. We are happy to include further ablations if requested.
>
>
> **Comment:** The presentation of the model needs a bit more clarification. The problem formulation can potentially be expanded a bit more (see questions below). The discussion of different training/inference choices is a bit dense. Maybe some concise pseudocode or flowchart in the main text could be helpful.
> **Response:** We follow the reviewer's suggestions. We have addressed the questions in the following and included a pseudocode of the unconditional generation in the main text. If the reviewer wishes, we will also include the algorithm of the conditional training or conditional prior sampling.
>
> **Comment:** If my understanding is correct, the overall idea of the model is to push an initial GP to the target time series. Therefore, the sequence length $L$ in the problem formulation section is equivalent to the vector field dimension $d$ in the background section. Can the author(s) kindly confirm if this is true? It would be nice to clarify the setting a little bit. I think the main source of confusion is that there are two time indices in this paper: the time in flow-matching and the time in the time series, and these are orthogonal to each other. I think this should be made clear somewhere in the paper.
> **Response:** The reviewer is correct that the sequence length $L$ corresponds to the vector field dimension $d$ in the background section. To address this, we have clarified the setting in the manuscript. Specifically, we now explicitly state that the flow-matching time $t$ and the time-series index are orthogonal, and we have updated the notation by using $\tau$ to represent the time indices within the time series time. Additionally, we have clarified that $L$ corresponds to $d$ in the background section.

---

> > ### Author Response · Authors · 2024-11-21
> >
> > **Comment:** Following the first question, it would be nice to indicate in Figure 1 the time-series index and the flow-matching index. Moreover, different notations should be used. (For instance, on line 202-203, $t$ should not be used for the kernel because it has already been used in Eq. (1).).
> > **Response:** We follow the reviewer's recommendation and have updated the indices and figure accordingly. Time indices within the time series are now denoted with $\tau$.
> >
> > **Comment:** How would the model respond to a growing $L$? What is the time complexity of the model and how does it compare to other models that you benchmarked against? When $L$ is large, Eq. (6) seems to suffer from the curse of dimensionality and the integral would be impossible to discretize. How does the model see this issue, or is it not relevant?
> > **Response:** Eq. (6) does not pose an issue for TSFlow, as there is no need to compute this integral directly. This integral, derived from Eq. (2), is standard in all CFM models. In practice, we sample from this distribution in a two-step process: (1) sample $\mathbf{x}_0 \sim q_0(\mathbf{x}_0 \mid \mathbf{y}^p)$; (2) solve the ODE with $\mathbf{x}_0$ as the initial condition.
> >
> >
> > The theoretical complexity of our prior distribution is $\mathcal{O}(L^3)$, where $L$ represents the length of the time series. This complexity arises from the Cholesky decomposition $(\mathbf{M}\mathbf{M}^T=\Sigma$, note that we used $\mathbf{M}$ instead of $\mathbf{L}$ to avoid ambiguity) needed to sample from the GP distribution in the unconditional setting, and from the inversion of the covariance matrix in Eq. (13) in the conditional one. However, these computations are performed only once, as the covariances remain fixed throughout training and sampling. Furthermore, $L$ is relatively small in our experiments (at most 360), making these computations minor.
> >
> >
> > During training, we can sample efficiently using a single matrix-vector multiplication. In the unconditional setting, this involves computing $\mathbf{x_0} = \mathbf{M}\mathbf{z}$, while in the conditional setting, we compute $\mu_{f|p} = \Sigma_{fp} \Sigma_{pp}^{-1} \mathbf{y}$ (see Eq. (13)). Both operations have a complexity of $\mathcal{O}(L^2)$, which is substantially lower than the initial setup.
> >
> >
> > To illustrate the scalability of TSFlow with longer sequences, we report the empirical runtimes per epoch (in seconds) for TSFlow and compare them to the diffusion-based baselines TSDiff and CSDI on the solar dataset:
> >
> > |Sequence length L | TSFlow (Isotropic - Uncond.)| TSFlow (OU - Uncond.) | TSFlow (OU - Cond.)|TSDiff|CSDI|
> > | -------- | -------- | -------- |-| - | - |
> > 360|3.674|3.743|3.508|3.787|3.629|
> > 720|4.004|4.115|3.582|4.190|9.760|
> > 1440|4.681|4.729|4.632|4.686|30.950|
> >
> > As shown in the table, the runtime difference between the isotropic and non-isotropic priors is minor. Note that CSDI is based on a transformer architecture, hence the increasing runtime for long sequences.
> >
> > **Comment:** Can you show some experiments where models that use the periodic kernel are indeed easier to train on tasks that involve periodicity?
> > **Response:** This can be observed in the LPS scores. While most datasets consist of a mixture of components such as periodicity, trends, and noise, making it challenging to classify them as strictly periodic, our results show that datasets with a dominant periodic component (e.g., solar, electricity, and traffic) achieve higher LPS scores when using a periodic prior (see Table 2). In contrast, the periodic prior performs worse on datasets like KDDCup, which are dominated by non-periodic components.
> >
> >
> > **Comment:** On line 302-303, you wrote "we additionally condition the prior distribution on the observed past data by approximating $q_0(\mathbf{x}_0\mid \mathbf{y}^p)$ with $q_0(\mathbf{x}_0\mid \mathbf{y}^p)$." I assume there is a typo. What would be the intended sentence?
> > **Response:** We thank the reviewer for spotting this typographical error. The correct phrase should be "factorizing $q_0(\mathbf{x}^p_0 \mid \mathbf{y}^p)$ with $q_0(\mathbf{x}^p_0 \mid \mathbf{y}^p)\,q_0(\mathbf{x}^f_0 \mid \mathbf{y}^p)$", as detailed in the "Gaussian Process Regression" paragraph. We have corrected this in the updated manuscript.
> >
> > We again thank the reviewer for their feedback and are happy to address any upcoming questions.
> >
> > [1] **Koochali, Alireza, et al.** "Random noise vs. state-of-the-art probabilistic forecasting methods: A case study on CRPS-Sum discrimination ability." Applied Sciences 12.10 (2022): 5104.
> >
> > [2] **Nie, Yuqi, et al.** "A Time Series is Worth 64 Words: Long-term Forecasting with Transformers." The Eleventh International Conference on Learning Representations.

---

> > > ### Comment · Reviewer_k8UA · 2024-11-26
> > >
> > > Thank you for your reply. There are a couple of lingering comments and concerns:
> > >
> > > 1. While you discussed channel-wise implementation to extend the model to the multivariate case, how would you handle the dependencies across different channels, if any? It would be useful to expand this discussion a bit in the manuscript and include some preliminary experiments. (At this stage of the rebuttal period, I am not expecting additional experimental results, but something to consider in the future.)
> > >
> > > 2. I understand that Eq. (6) is common in CFMs and does not need to be computed. My main confusion is that the integral in your algorithm includes an integration over an $L$-dimensional space. Even if (6) does not need to be computed, such a large dimension could make the discretization meaningless. That is, if you go to Lipman et al., then it is really hard to compute the expectation in Eq. (9) with only the sample. That would make the theoretical part questionable.
> > >
> > > I would maintain my position on the borderline.

---

> > > > ### Author Response · Authors · 2024-11-27
> > > >
> > > > We thank the reviewer for their response.
> > > >
> > > > **Regarding 1:** Incorporating dependencies across features in the multivariate case requires slight adjustments to our architecture. In the univariate setting, the input $\mathbf{x}_t$ has a shape of $(B, L, 1)$ and is transformed by the first convolution into $(B, L, H)$, where $H$ is the hidden dimension, and then reduced to $(B, L, 1)$ in the final convolution. For the multivariate case, the input $\mathbf{x}_t$ has a shape of $(B, L, K)$, where $K$ represents the number of channels, i.e., time series. This would require adapting the first convolution to process $(B, L, K)$ and adjusting the final convolution accordingly. This modification enables interaction between different channels.
> > > >
> > > > For the conditional feature vector $\mathbf{c}$, with a shape of $(B, L, C, K)$, where $C$ is the number of features (i.e., observation mask, observation, and lags), we would first flatten it to $(B, L, C \times K)$ before applying the convolution. We are currently working on multivariate experiments and will report the results as soon as they are complete.
> > > >
> > > > **Regarding 2:** We think there may be a misunderstanding. While our factorization in Eq. (6) contains the marginalization of $\mathbf{x}_0$ (or expectation over $\mathbf{x}_0$), which we wrote as an integral over an $L$-dimensional space, it is common practice to approximate such expectations with Monte Carlo samples during training and sampling. Furthermore, the dimension of our sample space, i.e., our sequence lengths, is far smaller than those of images.
> > > >
> > > > During training, we approximate the CFM objective (Eq. (9) in Lipman et al., Eq. (5) in our manuscript) across the entire batch. During inference, we approximate the cumulative distribution function of the forecast using 100 samples (see Evaluation Metrics in Section 4.).
> > > >
> > > > We hope this clarifies any misunderstandings.

---

> > > > > ### Author Response · Authors · 2024-12-02
> > > > >
> > > > > Dear Reviewer,
> > > > >
> > > > > We hope that the misunderstandings are clarified. In the meantime, we would like to share some initial results from our multivariate experiments. In the following table, we compare a channel-independent version of TSFlow (CIDP) to a multivariate extension (MV). The multivariate extension uses a transformer across the feature dimension as CSDI [1] to enable a fair comparison. As a reference, we also include the forecasting results (CRPS-Sum) of CSDI and Bilos et al. [2].
> > > > >
> > > > >
> > > > > | Model | Electricity | Exchange | Solar | Traffic |
> > > > > | -------- | -------- | -------- | ---- | --- |
> > > > > | Bilos et al. | 0.0260 $\pm$ 0.0011 | 0.0071 $\pm$ 0.0009 | 0.3701 $\pm$ 0.0179 | 1.2679 $\pm$ 1.0624|
> > > > > | CSDI | 0.0184 $\pm$ 0.0008 | 0.0076 $\pm$ 0.0022 | 0.3179 $\pm$ 0.0197 | 0.0221 $\pm$ 0.0012 |
> > > > > | TSFlow (CIDP) | 0.0196 $\pm$ 0.0006 | **0.0060** $\pm$ 0.0004 | 0.3780 $\pm$ 0.0074 | 0.0240 $\pm$ 0.0005 |
> > > > > | TSFlow (MV) | **0.0176** $\pm$ 0.0012 | 0.0073 $\pm$ 0.0005 | **0.2792** $\pm$ 0.0154 | **0.0217** $\pm$ 0.0013 |
> > > > >
> > > > >
> > > > > We are happy to address any remaining concerns.
> > > > >
> > > > > [1] **Tashiro, Yusuke, et al.** "Csdi: Conditional score-based diffusion models for probabilistic time series imputation." Advances in Neural Information Processing Systems 34 (2021): 24804-24816.
> > > > > [2] **Biloš, Marin, et al.** "Modeling temporal data as continuous functions with stochastic process diffusion." International Conference on Machine Learning. PMLR, 2023.

---

### Official Review · Reviewer_dfQz · 2024-10-23

**Soundness:** 3
**Presentation:** 2
**Contribution:** 2
**Rating:** 6
**Confidence:** 2

**Summary:**

This paper introduces TSFlow, a model for probabilistic time series forecasting that enhances generative modeling by incorporating Gaussian Process (GP) priors within the Conditional Flow Matching (CFM) framework. The use of more informative GP priors helps align the prior distribution with the temporal structure of the data, potentially improving both performance and leading to runtime efficiency by simplifying the probability paths. TSFlow is flexible and can be used for both, conditional and unconditional, time-series generation.

The model demonstrates strong performance across several benchmark datasets, but there are aspects of the paper that could benefit from further clarification and improvement. Depending on the author's response, I am willing to increase my score from "weak reject" to "weak accept".

**Strengths:**

**Incorporating Gaussian Process Priors:** The main contribution—replacing the typical isotropic Gaussian prior $q(x_0)$ with a data-dependent conditional prior $q(x_0∣y^p)$ is well-motivated. GP priors are naturally suited for time series due to their ability to model temporal dependencies, and this idea is a clear innovation over existing flow matching methods.

**Empirical Performance:** The empirical results show that TSFlow performs well compared to state-of-the-art models across various benchmark tasks.

**Flexibility in Conditional and Unconditional Modeling:** The approach supports both unconditional and conditional generation. While unconditional generation has fewer use cases compared to conditional generation, it is a feature often overlooked in time-series analysis. The ability to use the same model for both tasks—by applying conditioning only during inference—adds versatility to the paper.

**Weaknesses:**

**Majors**

**Difficulty in Parsing for Non-Experts:** The paper assumes substantial familiarity with flow matching and related generative methods, making it challenging for readers without a deep background in these specific techniques. It took for me considerable time to fully grasp the concepts, which suggests that the paper might also be difficult to read for a broader audience.

**Lack of Runtime Analysis:**  The authors propose replacing the isotropic Gaussian prior with a more complex GP prior. Howeer, they do not address the increased computational cost that comes with using GP priors. Although the GP prior is more suited to time-series tasks, this advantage must be weighed against the computational overhead. The paper should clearly state the theoretical runtime complexity of using GP priors and provide empirical runtime comparisons in the experiments. The reduction in NFEs is a positive, but its trade-off with the computational cost of the GP prior must be considered.

**Baseline Comparison:** A simple baseline for the forecasting tasks could involve using  Eq. 6 but with an isotropic Gaussian prior. Please include this method as a comparison partner in Table 3. If it is not a valid approach, it should be explained why such a baseline is excluded.

**Inconsistent Findings on Kernel Choice:** The periodic kernel minimizes the Wasserstein distance in Figure 2, suggesting it aligns well with the data distribution. However, in Table 1, the periodic kernel does not significantly outperform other kernels, and it is not even considered in Table 4.2. This inconsistency is counterintuitive and warrants further discussion. Moreover, the necessity for different hyperparameter choices across tasks (e.g., generative modeling vs. forecasting) weakens the "one model for all" argument, suggesting more task-specific tuning may be required.

**Minors:**

**Missing Experiments on Guided Generation (3.1.3):** I could not find experiments on Guided Generation (3.1.3) in the paper. Where can I find them?

**LPS Score Clarification:** The Linear Predictive Score (LPS) is not sufficiently explained. Please provide more details on how the LPS is calculated, specifically on what the linear model is regressed against and what the output represents.

**GP Hyperparameter Optimization:** You state in your paper that you did not fit the GP hyperparameters. This is somewhat unexpected, I would have had a strong guess that learning the hyperparameters of the GP brings a large benefit to the model. Can you perform a small experiment to evaluate the difference in performance with/without GP hyperparameter optimization?

**Questions:**

See above.

---

> ### Author Response · Authors · 2024-11-21
>
> We thank the reviewer for their valuable and comprehensive feedback. In the following, we address their concerns.
>
> **Comment:** Difficulty in Parsing for Non-Experts. The paper assumes substantial familiarity with flow matching and related generative methods, making it challenging for readers without a deep background in these specific techniques. It took for me considerable time to fully grasp the concepts, which suggests that the paper might also be difficult to read for a broader audience.
> **Response:** We thank the reviewer for the feedback. To make the paper easier to read, we have added more information to our methodology and included a pseudocode of the unconditional training in the main text. We are happy to include further information if requested.
>
> **Comment:** Lack of Runtime Analysis [...] The paper should clearly state the theoretical runtime complexity of using GP priors and provide empirical runtime comparisons in the experiments. [...].
> **Response:** The theoretical complexity of Gaussian Process priors is $\mathcal{O}(L^3)$, where $L$ denotes the length of the time series. In the unconditional setting, this complexity arises from the Cholesky decomposition $(\mathbf{M}\mathbf{M}^T = \Sigma$, note that we used $\mathbf{M}$ instead of $\mathbf{L}$ to avoid ambiguity) needed to sample from the GP distribution. In the conditional setting, it is due to the inversion of the covariance matrix in Eq. 13. However, both operations only need to be performed once, as the covariances are fixed and do not change throughout the sampling process. Furthermore, $L$ is relatively small in our experiments (at most 360), making these computations minor.
>
> During training, we can sample efficiently using matrix-vector multiplications. In the unconditional setting, this involves computing $\mathbf{x_0} = \mathbf{M} \mathbf{z}$ , while in the conditional setting, we compute $\mu_{f|p} = \Sigma_{fp} \Sigma_{pp}^{-1} \mathbf{y}$ (see Eq. 13). Both operations have a complexity of $\mathcal{O}(L^2)$, which is substantially lower than the initial setup.
>
> We include the runtimes in Appendix A.8 and, for convenience, also present them here. The table shows the training runtimes (seconds per epoch) and evaluation runtimes (seconds for generating all 100 forecasts on 959 time series) for the solar dataset.
>
> | Configuration                           | Train - Epoch | Eval. |
> |-----------------------------------------|---------------|-------|
> | TSFlow-Cond. – Isotropic prior          | 2.94          | 24.29 |
> | TSFlow-Cond. – GPR prior                | 3.17          | 24.65 |
> | TSFlow-Uncond. – Random                 | 3.66          | –     |
> | TSFlow-Uncond. – Optimal Transport + GP prior     | 3.77          | –     |
>
> As shown in the table, the computational overhead introduced by the non-isotropic priors is minimal, demonstrating that these priors do not substantially impact overall runtime.
>
> **Comment:** Baseline Comparison  A simple baseline for the forecasting tasks could involve using Eq. 6 but with an isotropic Gaussian prior. Please include this method as a comparison partner in Table 3. If it is not a valid approach, it should be explained why such a baseline is excluded.
> **Response:** We include a comparison to an isotropic Gaussian prior in Appendix B.1.1. (see *Basemodel* in Table 7). If the reviewer wishes, we will include this baseline in Table 3.
>
> **Comment:** Inconsistent Findings on Kernel Choice The periodic kernel minimizes the Wasserstein distance in Figure 2, suggesting it aligns well with the data distribution. However, in Table 1, the periodic kernel does not significantly outperform other kernels [...]. This inconsistency is counterintuitive and warrants further discussion. Moreover, the necessity for different hyperparameter choices across tasks (e.g., generative modeling vs. forecasting) weakens the "one model for all" argument, suggesting more task-specific tuning may be required.
> **Response:** We would like to clarify that to parametrize the Gaussian processes, we chose $\ell=\sqrt{\frac{1}{2}}$, $\ell=1$, and $\ell=\sqrt{2}$, for the SE, OU, and PE kernels, respectively, for all experiments. The reason for this is that they simplify the kernels even further to: $K_{\mathrm{SE}}(\tau, \tau') = \exp(-d^2)$, $K_{\mathrm{OU}}(\tau, \tau') = \exp(-\mid d\mid)$, and $K_{\mathrm{PE}}(\tau, \tau') = \exp(\sin^2(d))$, which are used for all experiments, including generative modeling and forecasting.
>
> Furthermore, the periodic kernel outperforms other methods in downstream tasks, as quantified by the LPS, consistent with the insights presented in Figure 2.

---

> ### Author Response · Authors · 2024-11-21
>
> **Comment:** Missing Experiments on Guided Generation (3.1.3): I could not find experiments on Guided Generation (3.1.3) in the paper. Where can I find them?
> **Response:** The results are shown in Table 3 under the name *TSFlow-Uncond*. We apologize for the confusion and have clarified this in the text. Additionally, we show the results of the guided generation without conditional prior sampling in Table 8 in Appendix B.1.2 and ablate the guidance score function in Appendix B.1.3.
>
> **Comment:** LPS Score Clarification: The Linear Predictive Score (LPS) is not sufficiently explained. Please provide more details on how the LPS is calculated, specifically on what the linear model is regressed against and what the output represents.
> **Response:** The Linear Predictive Score (LPS) is computed as follows: First, synthetic samples $\mathbf{y}=(\mathbf{y}^p,\mathbf{y}^f)$ are drawn from the generative model. A linear regression model is then trained to map the past of the time series ($\mathbf{y}^p$) to its future values ($\mathbf{y}^f$), using the true future values for supervision. Since it is a linear regression model, the parameters can be computed in closed form without requiring iterative optimization. Finally, the LPS is then computed as the CRPS of the linear model's forecasts on the real test set. We have provided additional details in Appendix A.5 and added a reference to this explanation in the main text.
>
> **Comment:** GP Hyperparameter Optimization: Can you perform a small experiment to evaluate the difference in performance with/without GP hyperparameter optimization?
> **Response:** We followed the reviewer's recommendation and conducted the requested experiment. For this evaluation, we used the periodic kernel, parameterized with a learnable parameter $\gamma$: $K_{\mathrm{PE}}(\tau, \tau') = \exp(\gamma\sin^2(d))$
>
> We initialized $\gamma=1.0$ and optimized it by maximizing the likelihood using 10,000 samples from the real datasets. The optimization was performed for 1,000 iterations with Adam. In the following table, we report the $W_2^2$ distances of the generative model trained with the priors under both fixed and fitted values of $\gamma$.
>
>
> **4 NFEs:**
> | $\gamma$ | Electricity | KDDCup | M4 (H) | Solar | Traffic |
> | - | - | -| -| -| -|
> | 1.0|  **2.501** $\pm$ 0.091 | 25.712 $\pm$ 0.437 | **6.353**  $\pm$ 0.184 | 4.456 $\pm$ 0.140 |  7.411  $\pm$ 0.210 |
> | Fitted | 2.684 $\pm$ 0.119  |   **25.647** $\pm$ 0.810 | 6.596 $\pm$ 0.707 | **4.312** $\pm$ 0.091 | **7.115** $\pm$ 0.091 |
>
> **16 NFEs:**
> | $\gamma$ | Electricity | KDDCup | M4 (H) | Solar | Traffic |
> | - | - | -| -| -| -|
> | 1.0 | **2.532** $\pm$ 0.056 | 26.501 $\pm$ 0.920 | **6.158** $\pm$ 0.181 | 4.504 $\pm$ 0.091 | 7.202 $\pm$ 0.097 |
> | Fitted | 2.584 $\pm$ 0.150  |   **25.815** $\pm$ 0.372 | 6.395 $\pm$ 0.484 | **4.483** $\pm$ 0.013 | **7.109** $\pm$ 0.153 |
>
> Across these experiments, we observe an improvement in 6 out of 10 cases when the kernel parameter $\gamma$ is optimized. Thus, while hyperparameter optimization can lead to better performance in some settings, it is not necessary for the generative model.
>
>
> Again, we thank the reviewer for their feedback and are happy to address any remaining concerns.

---

> > ### Comment · Reviewer_dfQz · 2024-11-23
> >
> > Thanks for your reply. I am satisfied with all the answers you provided except of the answer to "Inconsistent Findings on Kernel Choice" where I think we misunderstood each other.
> >
> > I also understood that you used universal parameterizations for the different kernels. My comment related to the kernel choice (and not the parameterization choice). You wrote in the paper, and also in your initial response, that the periodic kernel performs best on downstream tasks which is consistent with the insights presented in Figure 2. Yet, I didn't find the periodic kernel in Table 4.2, but only the OU kernel.
> >
> > Can you add the numbers of the other kernels to Table 4.2 or at least provide a high-level recommendation what kernel to use if I want to use the same model for both, unconditional and conditional, tasks? At the moment, I just know that the periodic kernel works best for unconditional tasks and the OU kernel for conditional tasks. But I do not know which kernel to use if I want to have one model for both tasks. If no model works good for both tasks, this has to be at least stated clearly as a limitation.

---

### Official Review · Reviewer_jGDf · 2024-11-01

**Soundness:** 4
**Presentation:** 4
**Contribution:** 2
**Rating:** 8
**Confidence:** 4

**Summary:**

This paper studies the use of flow matching for time series forecasting. The authors first propose the use of flow matching techniques for unconditional generation using Gaussian process priors (section 3.1.1), followed by a technique for conditioning an unconditional model by sampling a relevant prior $x_0$ (section 3.1.2) or guidance (section 3.1.3), and finally the authors discuss a technique which uses a data-dependent Gaussian process prior for conditional sampling (section 3.2). The proposed methodology is empirically validated on several univariate time series datasets.

**Strengths:**

- The investigation of techniques for conditional sampling is pretty thorough, and I think the proposed methods are quite interesting
- The proposed method obtains fairly strong empirical results, and the empirical evaluation is convincing
- Throughout the paper is very clear and well-written

**Weaknesses:**

- There is some highly relevant related work that the authors do not discuss. [Functional Flow Matching, AISTATS 2024](https://arxiv.org/abs/2305.17209) proposes the use of GP priors in conjunction with flow matching and studies techniques for forecasting with these models.  Similarly, [Conditional Flow Matching for Time Series Modeling, SPIGM@ICML 2024](https://openreview.net/forum?id=Hqn4Aj7xrQ) uses GPs with flow matching for time series. The authors should cite these works and discuss the differences with their proposed method.
- There are some (relatively minor) clarity issues throughout
     - The use of equation 9 was a bit unclear to me. Why is this specific form of $q_1$ chosen? Some justification for this modeling choice would be good.
     - In Section 3.1.2, I am guessing that once we sample $x_0 \sim q_0(x_0 \mid y^p)$, then we use $x_0$ as an initial condition for the flow model to generate new samples $y \mid x_0$. Is this the case? If so, it would help to state this explicitly somewhere in the paper.
     - In Line 303, the authors write "approximating q_0(x_0 \mid y^p)$ with $q_0(x_0 \mid y^p)$. I think I understand what is meant here, but this seems to be a typo.

**Questions:**

- The exact problem statement was a bit unclear to me. In lines 154-156 the authors describe a time series as a vector in $\mathbb{R}^L$. Does this mean that the authors only work on time series having a fixed length $L$, or does the method allow for variable-length time series? Similarly, are the authors assuming that the time series all share a fixed discretization (i.e., there are some fixed times $t_1, \dots, t_L$ corresponding to $y_1, \dots, y_L$), or can the discretization vary across time series? Is this discretization assumed to be uniform, or can it be irregular as well?
- It seems to me that the setup is not limited just to forecasting, but could be applied to general conditional generation tasks, e.g., imputation. Have the authors tried anything beyond forecasting?

---

> ### Author Response · Authors · 2024-11-21
>
> We thank the reviewer for their valuable feedback. In the following, we address their comments and questions.
>
>
> **Comment:** There are missing related works  [8,9].
> **Response:** We thank the reviewer for pointing out the references. We have included them in our related work section.
>
> While there are multiple differences to [8], the most notable one is that [8] models a dynamical system using CFM, where the time series and ODE share the time dimension. In contrast, TSFlow introduces two orthogonal time dimensions: one along the generative process and another within the time series itself. This allows TSFlow to employ a sequence-to-sequence model across all generation steps.
>
> As for [9], it extends the flow-matching framework to infinite-dimensional spaces and also leverages Gaussian processes. However, their key-focus is different. More specifically, [9] constructs conditional Gaussian measure paths, whereas TSFlow focuses on parameterizing the joint distribution aligning with its sequence modeling framework. This leads to multiple differences, such as our Gaussian process regression and conditional prior sampling or the choice of our architecture.
>
>
> **Comment:** The use of equation 9 was a bit unclear to me. Why is this specific form of $q_1$ chosen?
> **Response:** We use this specific form of equation 9 due to its connection with Bayesian quantile regression [1]. Applying the log-likelihood (see Eq. 8) leads to the quantile loss. The quantile loss is a CDF-based score and common practice in probabilistic forecasting [2, 3, 4].
>
> We have added a section in Appendix A.3 discussing this choice. Furthermore, we have added an experiment in Appendix B.2, comparing it to a Gaussian distribution, which yields inferior results.
>
> **Comment:** In Section 3.1.2, I am guessing that once we sample $x_0 \sim q_0(x_0∣y^p)$, then we use $x_0$ as an initial condition for the flow model to generate new samples $y\mid x_0$. Is this the case? If so, it would help to state this explicitly somewhere in the paper.
> **Response:** The reviewer is correct that conditional prior sampling involves a two-step process. We have clarified this procedure in Section 3.1.2 to make the sampling steps more explicit.
>
> **Comment:** In Line 303, the authors write "approximating $q_0(x_0 \mid y^p)$ with $q_0(x_0 \mid y^p)$". I think I understand what is meant here, but this seems to be a typo.
> **Response:** The reviewer is correct. The correct phrase should be "factorizing $q_0(\mathbf{x}^p_0 \mid \mathbf{y}^p)$ with $q_0(\mathbf{x}^p_0 \mid \mathbf{y}^p)\,q_0(\mathbf{x}^f_0 \mid \mathbf{y}^p)$", as detailed in the "Gaussian Process Regression" paragraph. We have corrected this typo.
>
> **Comment:** In lines 154-156 the authors describe a time series as a vector in $\mathbb{R}^L$. Does this mean that the authors only work on time series having a fixed length $L$, or does the method allow for variable-length time series? Similarly, are the authors assuming that the time series all share a fixed discretization (i.e., there are some fixed times $(t_1,\dots,t_L)$ corresponding to $(y_1,\dots,y_L)$, or can the discretization vary across time series? Is this discretization assumed to be uniform, or can it be irregular as well?
> **Response:** Our experiments considered time series of fixed length $L$ with fixed uniform discretizations in hourly and daily intervals (see Tab. 4 in the appendix). This is consistent with previous works [4,5,6]. In theory, TSFlow can be trained on time series of varying lengths $L$. Applying TSFlow to irregular time series would require a slight adaption of the network architecture. More specifically, one would need to include positional embeddings of the time steps as input to the neural network, as done in [7]. We have clarified in the paper that we use regularly sampled time series.

---

> ### Author Response · Authors · 2024-11-21
>
> **Comment:** Could the model be applied to general conditional generation tasks, e.g., imputation. Have the authors tried anything beyond forecasting?
> **Response:** Our methodology is not technically limited to forecasting and can be adapted to general conditional generation tasks, including imputation. However, we chose to focus on forecasting in our experiments due to the broader availability of standardized benchmarks and to ensure fair comparisons with autoregressive models, which do not benefit from bidirectional information passing.
>
> We conducted a *forecasting with missing values* experiment to explore applications beyond standard forecasting. Following the setup in [4], we evaluated three scenarios: random missing values, blackout missing at the beginning of the context window, and blackout missing at the end of the context window. We masked 50% of the values in each scenario and assessed the forecasting performance using CRPS. Below, we report results for TSFlow (Cond.) and TSDiff. For TSDiff, we report the minimum performance across both versions.
>
>
> Random Missing:
>
> | Dataset | Electricity | Exchange | KDDCup | Solar | Traffic | UberTLC |
> | -------- | -------- | -------- | - | - | -| -|
> | TSFlow | **0.048** $\pm$ 0.001 | **0.009** $\pm$ 0.001 | 0.432 $\pm$ 0.121 | 0.362 $\pm$ 0.006 | **0.085** $\pm$ 0.000 | **0.157** $\pm$ 0.002 |
> | TSDiff | 0.052 $\pm$ 0.001 | 0.012 $\pm$ 0.004 | **0.397** $\pm$ 0.042| **0.357** $\pm$ 0.023 | 0.097 $\pm$ 0.003 | 0.180 $\pm$ 0.015 |
>
> Blackout Missing Beginning:
>
> | Dataset | Electricity | Exchange | KDDCup | Solar | Traffic | UberTLC |
> | -------- | -------- | -------- | - | - | -| -|
> | TSFlow | **0.047** $\pm$ 0.001 | **0.008** $\pm$ 0.001 | **0.305** $\pm$ 0.013 | 0.396 $\pm$ 0.026 | **0.083** $\pm$ 0.000 | **0.158** $\pm$ 0.008 |
> | TSDiff | 0.049 $\pm$ 0.001 | 0.009 $\pm$ 0.000 | 0.441 $\pm$ 0.096 | **0.377** $\pm$ 0.017 | 0.094 $\pm$ 0.005 |0.181 $\pm$ 0.009 |
>
> Blackout Missing End:
>
> | Dataset | Electricity | Exchange | KDDCup | Solar | Traffic | UberTLC |
> | -------- | -------- | -------- | - | - | -| -|
> | TSFlow | **0.061** $\pm$  0.001 | 0.056 $\pm$ 0.004 | 0.384 $\pm$ 0.033 | 0.418 $\pm$ 0.014 | **0.106** $\pm$ 0.002 | **0.165** $\pm$ 0.002 |
> | TSDiff |  0.065 $\pm$ 0.003 | **0.020** $\pm$ 0.001 | **0.344** $\pm$ 0.012 | **0.376** $\pm$ 0.036 | 0.123 $\pm$ 0.023 | 0.179 $\pm$ 0.013 |
>
> Overall, TSFlow outperforms both versions of TSDiff in 12 out of 18 cases, demonstrating its robustness across these scenarios. If the reviewer wishes, we will include these results in the paper.
>
>
> We again thank the reviewer for their feedback and are happy to address any remaining concerns.
>
> [1] **Yu, Keming, and Rana A. Moyeed.** "Bayesian quantile regression." Statistics & Probability Letters 54.4 (2001): 437-447.
> [2] **Lim, Bryan, et al.** "Temporal fusion transformers for interpretable multi-horizon time series forecasting." International Journal of Forecasting 37.4 (2021): 1748-1764.
> [3] **Gouttes, Adèle, et al.** "Probabilistic time series forecasting with implicit quantile networks." arXiv preprint arXiv:2107.03743 (2021).
> [4] **Kollovieh, Marcel, et al.** "Predict, refine, synthesize: Self-guiding diffusion models for probabilistic time series forecasting." Advances in Neural Information Processing Systems 36 (2024).
> [5] **Tashiro, Yusuke, et al.** "Csdi: Conditional score-based diffusion models for probabilistic time series imputation." Advances in Neural Information Processing Systems 34 (2021): 24804-24816.
> [6] **Alcaraz, Juan Lopez, and Nils Strodthoff.** "Diffusion-based Time Series Imputation and Forecasting with Structured State Space Models." Transactions on Machine Learning Research.
> [7] **Biloš, Marin, et al.** "Modeling temporal data as continuous functions with stochastic process diffusion." International Conference on Machine Learning. PMLR, 2023.
> [8] **Tamir, Ella, et al.** "Conditional flow matching for time series modelling." ICML 2024 Workshop on Structured Probabilistic Inference {\&} Generative Modeling. 2024.
> [9] **Kerrigan, Gavin, Giosue Migliorini, and Padhraic Smyth.** "Functional flow matching." arXiv preprint arXiv:2305.17209 (2023).

---

> > ### Comment · Reviewer_jGDf · 2024-11-27
> >
> > Thanks so much for this clear and constructive response. I appreciate the additional clarifications the authors have added to the paper -- these generally make the approach very clear to me now. Overall I quite like the paper and I'm happy to increase my score given these changes.
> >
> > > If the reviewer wishes, we will include these results in the paper.
> >
> > I think your point regarding autoregressive models, i.e. that they "do not benefit from bidirectional information passing," is a convincing reason for focusing on forecasting. Perhaps a sentence or two alluding to this (i.e., that your model can perform other tasks, but you focus on forecasting) could be useful for other readers with the same idea.
> >
> > While I think these additional experiments could serve to strengthen the paper, they might require re-framing the writing a bit to emphasize that it is not just forecasting the model is capable of, and finding additional baselines. I'm happy to leave this decision up to the authors.

---

> > > ### Author Response · Authors · 2024-11-27
> > >
> > > We thank the reviewer for their response and for adjusting their score.
> > >
> > > We are pleased that the reviewer is satisfied with our response and appreciates the paper. The results have been included in the updated manuscript in Appendix B.3.
> > >
> > > We are happy to include any additional feedback to improve the paper further.

---

### Official Review · Reviewer_qDmP · 2024-11-04

**Soundness:** 3
**Presentation:** 3
**Contribution:** 3
**Rating:** 8
**Confidence:** 2

**Summary:**

The existing diffusion models have problems in the time series generation since the data and prior distributions differ. The authors handle this problem by utilizing conditional flow matching framework. They propose TSFlow, which sets the prior distribution as Gaussian process to make the prior distribution close to the data distribution. Also, they propose conditional prior sampling which makes an unconditionally trained model possible for probabilistic forecasting.

**Strengths:**

• By utilizing Gaussian process to the conditional flow matching, the model reflects the temporal dependencies of the given time series data better.

• The model enables both unconditional and conditional generations.

• By conditional prior sampling, the unconditionally trained model could follow the given guidance.

**Weaknesses:**

Please refer to the Questions section.

**Questions:**

•	The problem only considers the univariate case. Can the model extend to the multivariate time series problem?

•	I want some more explanation about the effectiveness of informed prior distributions. Why does closedness of the prior and data distribution imply easy learning. Do you have any experiments about train efficiency or path efficiency?

•	Can the given prior (Gaussian process) extend to the arbitrary prior ? for example, refer to [1].

•	Do you have any theoretical evidence about how the selection of kernel functions effect to the model performance? (ex. OU kernel is better when the data follows OU process)

[1] Leveraging Priors via Diffusion Bridge for Time Series Generation, Arxiv24

---

> ### Author Response · Authors · 2024-11-21
>
> We thank the reviewer for their valuable feedback and comments. In the following, we address their remarks.
>
> **Comment:** Can the model extend to the multivariate time series problem?
> **Response:** The model can be extended to the multivariate time series problem by adjusting the neural network $u_\theta(t,x_t)$ to multivariate time series, i.e., $u_\theta(t,x_t):[0,1]\times\mathbb{R}^{K\times L}\to\mathbb{R}^{K\times L}$.
>
> To extend the neural network, one could use a channel-independent architecture such as PatchTST [1] or employ a feature interaction layer (e.g. transformer) to process the multivariate time series.
>
> We have decided to perform experiments on univariate data as these present many real-world problems. Furthermore, metrics such as the CRPS-sum specifically designed for multivariate forecasting have several limitations, particularly when evaluating individual dimensions' performance as shown in [2].
>
>
> **Comment:** Why does closedness of the prior and data distribution imply easy learning. Do you have any experiments about train efficiency or path efficiency?
> **Response:** Non-isotropic priors introduce patterns that naturally occur in real data, thereby preserving the non-i.i.d. characteristics during the noising/forward process. This introduces an inductive bias throughout the entire generation process. In contrast, the isotropic prior starts with initial samples without interdependencies, requiring the model to learn and introduce these during generation.
>
>
> We have included Figure 7, which shows the Wasserstein distances of the synthetic samples across different numbers of neural function evaluations (NFEs). The non-isotropic priors demonstrate faster convergence compared to the isotropic prior.
>
> During training, we observed that the non-isotropic priors reached the performance levels of the isotropic prior faster. More specifically, to achieve the best Wasserstein distances (at 16 NFEs) obtained by the isotropic prior, the non-isotropic priors reduce the number of training iterations by nearly 50%. The table below summarizes the percentage of training iterations required by each non-isotropic prior, relative to the isotropic prior, to reach this performance.
>
> | Kernel | Amount of training iterations |
> | -------- | -------- |
> | Ernstein Uhlenbeck     | 54.23%     |
> | Periodic | 52.82% |
> | Squared Exponential| 59.15% |
>
> Note that we excluded the few cases where the isotropic prior outperformed the non-isotropic priors.
>
> **Comment:** Can the given prior (Gaussian process) extend to the arbitrary prior?
> **Response:** We thank the reviewer for pointing us to this reference. Yes, the prior in TSFlow can be extended to arbitrary source distributions. Importantly, we do not require access to the PDF or CDF of the distribution; we only need to be able to sample from it. The adaptation only involves modifying the distribution $q_0$, while $q_1$, $p_t$, and $u_t$ remain unchanged.
>
> We included this in the updated manuscript and cited the reference.
>
> **Comment:** Do you have any theoretical evidence about how the selection of kernel functions effect to the model performance?
> **Response:** Since most datasets consist of a mixture of components, such as periodicity, trends, and noise, it is challenging to assign specific kernels to datasets definitively. However, our results indicate that datasets with a dominant periodic component (e.g., solar, electricity, and traffic) achieve better LPS scores when using a periodic prior (see Table 2). Conversely, on datasets like KDDCup, which are dominated by non-periodic components, the periodic prior struggles to deliver strong performance.
>
> We hope that the questions are answered satisfactorily and are happy to address any upcoming concerns.
>
> [1] **Nie, Yuqi, et al.** "A Time Series is Worth 64 Words: Long-term Forecasting with Transformers." The Eleventh International Conference on Learning Representations.
>
> [2] **Koochali, Alireza, et al.** "Random noise vs. state-of-the-art probabilistic forecasting methods: A case study on CRPS-Sum discrimination ability." Applied Sciences 12.10 (2022): 5104.

---

> > ### Comment · Reviewer_qDmP · 2024-11-25
> >
> > Many thanks for your response.  I have decided to increase my score although I still feel that your responses do not fully address my concerns. I want to see the exact experimental methods such as how to extend to the multivariate case, how to sample from non-Gaussian prior, and how to select the kernel functions and corresponding effects. I know that the time is not enough to perform all these experiments. Since I intuitionally agree your responses, I increase my score. But I think if the experiments are added, your paper will be more theoretically robust paper.

---

> > > ### Author Response · Authors · 2024-11-25
> > >
> > > We thank the reviewer for their response and adjustment of the score.
> > >
> > > We agree that experiments including the multivariate case and non-Gaussian priors would enhance the paper. While time constraints prevent us from addressing these fully now, we will include them in the final version.

---

### Meta-Review · Area_Chair_FwWH · 2024-12-21

**Metareview:**

This paper introduces a new generative framework for time series forecasting, based on conditional flow matching and leveraging conditional Gaussian Processes as informed priors. By aligning the prior distribution more closely with the data distribution, this approach simplifies the probability paths. In turn, the experimental results demonstrate that this approach helps to improve the generative performance for forecasting.

Overall, the reviewers are highly positive about this work. They emphasize its originality, strong empirical results, and the intuitive nature of the informed prior sampling strategy. The paper is also well written and clearly presented. One limitation is that the approach has primarily been validated on univariate time series problems.

Given the importance of machine learning for time series, this paper makes a valuable contribution. The positive and supportive reviews underline its merit, and I feel that this paper will motivate future work. I therefore recommend accepting this paper.

**Additional Comments On Reviewer Discussion:**

The authors provided a detailed rebuttal that addressed the reviewers’ concerns, including an extended ablation study, a revised manuscript clarifying several key points, and an expanded related work section.

---

### Decision · Program_Chairs · 2025-01-22

Accept (Poster)